# FedVLMBench: Benchmarking Federated Fine-Tuning of Vision-Language Models

**Weiying Zheng**[1] **Ziyue Lin**[1] **Pengxin Guo**[1] **Yuyin Zhou**[2] **Feifei Wang**[1] **Liangqiong Qu**[1]

[1]The University of Hong Kong    [2]UC Santa Cruz

## Abstract

Vision-Language Models (VLMs) have demonstrated remarkable capabilities in cross-modal understanding and generation by integrating visual and textual information. While instruction tuning and parameter-efficient fine-tuning methods have substantially improved the generalization of VLMs, most existing approaches rely on centralized training, posing challenges for deployment in domains with strict privacy requirements like healthcare. Recent efforts have introduced Federated Learning (FL) into VLM fine-tuning to address these privacy concerns, yet comprehensive benchmarks for evaluating federated fine-tuning strategies, model architectures, and task generalization remain lacking. In this work, we present **FedVLMBench**, the first systematic benchmark for federated fine-tuning of VLMs. FedVLMBench integrates two mainstream VLM architectures (encoder-based and encoder-free), four fine-tuning strategies, five FL algorithms, six multimodal datasets spanning four cross-domain single-task scenarios and two cross-domain multitask settings, covering four distinct downstream task categories. Through extensive experiments, we uncover key insights into the interplay between VLM architectures, fine-tuning strategies, data heterogeneity, and multi-task federated optimization. Notably, we find that a 2-layer multilayer perceptron (MLP) connector with concurrent connector and LLM tuning emerges as the optimal configuration for encoder-based VLMs in FL. Furthermore, current FL methods exhibit significantly higher sensitivity to data heterogeneity in vision-centric tasks than text-centric ones, across both encoder-free and encoder-based VLM architectures. Our benchmark provides essential tools, datasets, and empirical guidance for the research community, offering a standardized platform to advance privacy-preserving, federated training of multimodal foundation models. Our dataset and code are publicly available.

## 1 Introduction

Recently, Vision-Language Models (VLMs) [1, 20, 31] have demonstrated groundbreaking advancements in cross-modal understanding and generation tasks by integrating multimodal information such as vision and language. Instruction tuning methods, such as LLaMA-Adapter V2 [5], and parameter-efficient tuning techniques, such as LoRA [10], can significantly enhance the zero-shot generalization capabilities of VLMs. This characteristic positions VLMs as a potential foundational architecture for addressing complex open-domain tasks. However, existing VLM-based instruction tuning methods [5, 10, 20] typically adopt a centralized learning paradigm, which fails to meet the privacy protection requirements necessary for distributed training, particularly in sensitive fields such as healthcare. While recent research [34, 40] has introduced FL into the instruction fine-tuning of VLMs to effectively address data privacy concerns, significant limitations remain.

First, existing VLMs can be categorized into two popular technical routes, encoder-based VLMs and encoder-free VLMs, depending on the inclusion of visual encoders [30, 32]. Current methods

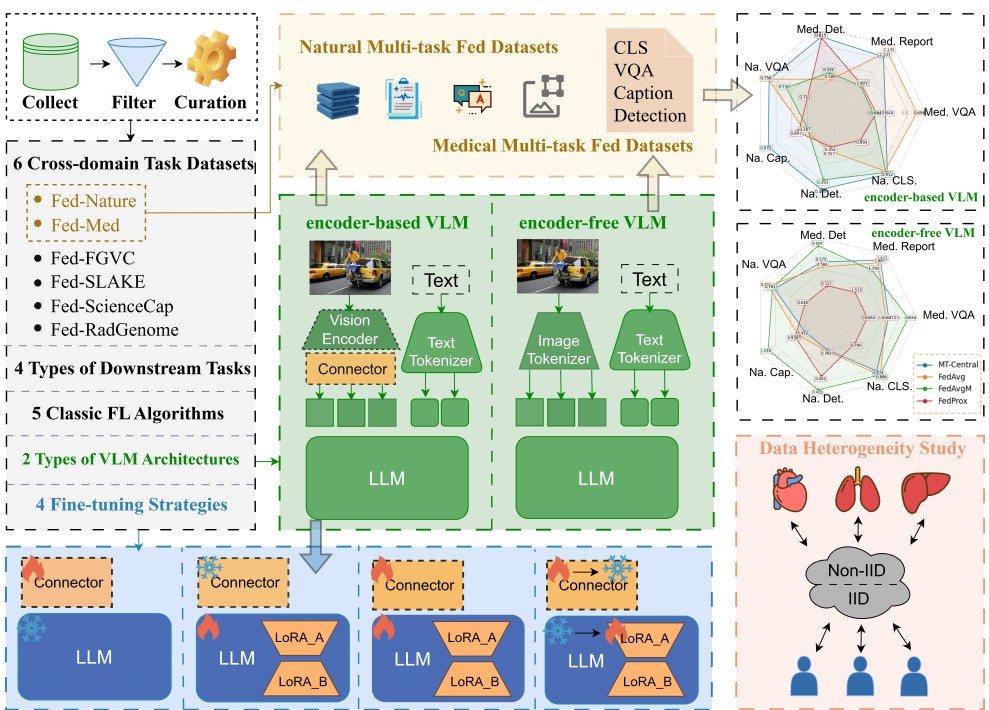

Figure 1: Overview of our proposed FedVLMBench, which integrates two types of mainstream VLM architectures, four fine-tuning strategies, five FL algorithms, and six cross-domain task datasets. This framework facilitates comprehensive evaluation and comparison of multitask learning approaches in FL contexts.

primarily focus on adapting encoder-based VLMs through techniques such as LoRA or global fine-tuning, lacking a systematic comparison framework and benchmark for different model architectures and fine-tuning strategies. Second, existing FL multimodal benchmark research focuses narrowly on two basic task types—Visual Question Answering (VQA) and classification—while ignoring more complex but critically important multimodal tasks such as report generation and visual localization (Tab.1). Third, no existing FL datasets support federated multi-modal multi-task learning scenarios, despite their practical significance in real-world applications where different clients may need to handle distinct multimodal tasks (e.g., one hospital specializes in classification while another focuses on report generation). To address these research gaps, this paper shifts from technical improvements in existing federated instruction tuning methods to exploring three core foundational questions:

**Q1**: How do choices in connector design and fine-tuning strategies impact the FL performance of encoder-based VLMs across diverse single-learning FL tasks?

**Q2**: How do different FL algorithms, using both encoder-based and encoder-free VLMs as baseline architectures, perform under varying data heterogeneity conditions in single-task federated fine-tuning processes?

**Q3**: To what extent can existing FL algorithms support multi-modal multi-task coordination when deploying heterogeneous VLMs across clients with divergent task requirements?

To systematically address these questions, we developed an innovative FL fine-tuning of VLMs benchmark **FedVLMBench** that integrates **2** types of mainstream VLM architectures (encoder-based and encoder-free VLMs), **4** fine-tuning strategies, **5** FL algorithms, **4** types of downstream tasks, and **6** cross-domain task datasets. As shown in Tab.1, our benchmark differs from existing works by encompassing a broader range of downstream tasks, diverse VLM architectures, and unique multi-task collaborative fine-tuning datasets. Through extensive experimental analysis, we present the following key findings:

Table 1: Comparisons of FedVLMBench with other FL benchmarks.#Collab. Datasets refer to the number of multi-task collaborative fine-tuning datasets.

| Benchmark | Language | Vision | # Arch. Types | # Task Types | # Datasets | # Collab. Datasets |
|-----------|----------|--------|---------------|--------------|------------|--------------------|
| FS-LLM [14] | ✓ | ✗ | 1 | 2 | 3 | ✗ |
| FedLLM-Bench [36] | ✓ | ✗ | 1 | 2 | 4 | ✗ |
| OpenFedLLM [37] | ✓ | ✗ | 1 | 2 | 8 | ✗ |
| FedMLLM [34] | ✓ | ✓ | 1 | 2 | 5 | ✗ |
| **FedVLMBench** (Ours) | ✓ | ✓ | **2** | **4** | 6 | **2** |

1) For encoder-based VLM in FL, a 2-layer MLP connector stands out as the most effective connector when compared to other linear or more complex MLP configurations; concurrent fine-tuning both the connector and the LLM yields superior task-agnostic performance compared to the sequential approach of fine-tuning the connector first and then the LLM, while maintaining computational efficiency.

2) For encoder-based VLMs in FL, text-centric tasks (such as VQA and caption generation) benefit dominantly from LLM fine-tuning, while connector fine-tuning should be prioritized for vision-centric tasks like classification and detection.

3) Current FL optimization methods are ineffective for both encoder-free and encoder-based VLMs when dealing with non-IID data partitions in single-task FL learning, calling for novel solutions addressing vision-centric heterogeneity challenges.

4) While single-task FL struggles with vision-centric performance degradation under non-IID data, federated multitask training achieves near-ceiling performance comparable to centralized training across both text- and vision-centric tasks, regardless of VLM architectures.

The main contributions of this paper can be summarized as follows:

1. We propose **FedVLMBench**, the first systematic benchmark for federated fine-tuning of VLMs. It integrates two mainstream VLM architectures (encoder-based and encoder-free), four fine-tuning strategies, five diverse FL algorithms, and six cross-domain datasets spanning task categories from text-centric (VQA/captioning) to vision-intensive (classification/detection), while comprehensively supporting both single-task and multi-task FL scenarios.

2. We bridge critical gaps in FL benchmarks by introducing (i) four cross-domain single-task datasets with configurable IID, simulated non-IID, and real-world non-IID data distributions, and (ii) two novel multi-task vision-language datasets reflecting real-world non-IID scenarios where clients handle distinct yet interconnected tasks.

3. Through comprehensive evaluation on FedVLMBench, we establish actionable guidelines for federated fine-tuning of VLMs and reveal open challenges for future research in privacy-preserving FL multimodal systems.

## 2 Related Work

**Vision-Language Models** (VLMs) [1, 31] have rapidly advanced by significantly enhancing perceptual and reasoning capabilities through the integration of multimodal information, including text, images, and video. Currently, VLMs can be categorized into two primary types: encoder-based models and encoder-free models. The former encompasses models such as LLAVA [20], which utilize pretrained encoders (e.g., CLIP [25]) to extract multimodal features and integrate them with LLMs for executing complex tasks. In contrast, encoder-free models [17, 32] directly tokenize multimodal data, such as images, enabling adaptive processing of diverse inputs and enhancing the generalizability of VLMs.

**Federated Learning** (FL) [6, 7, 8, 23, 38, 41] is a privacy-preserving distributed training paradigm that facilitates collaborative modeling through client-localized data processing. The traditional FedAvg [23] method relies on client data volume for parameter-weighted fusion but often suffers from performance degradation in non-IID scenarios. To address this, various optimization schemes have been proposed, such as FedProx [16], FedAdagrad [27], FedAdam, and FedYogi [28], PerAvg [4], and FedTGP [39]. More recently, researchers have begun exploring FL in the context of multimodal learning, such as FedLPS [11], FedMBridge [2], and Pilot [33]. For example, Pilot [33] tackles the

Table 2: Statistics of 6 federated multimodal fine-tuning datasets in FedVLMBench.

| Dataset | Task Type | Data Source | Data Type | #Max Clients | #Instances | Evaluate metric |
|---|---|---|---|---|---|---|
| Fed-FGVC | CLS | FGVC [22] | Image | 30 | 9,967 | Acc |
| Fed-ScienceCap | Caption Generation | ScienceQA [21] | Image+Text | 27 | 5,157 | CIDER/ROUGE_L |
| Fed-SLAKE | VQA | SLAKE [19] | Image+Text | 3 | 8,061 | Acc |
| Fed-RadGenome | Detection | RadGenome-Chest CT [42] | Image+Text | 3 | 8,744 | IoU |
| Fed-Nature | VQA | COCO-QA [29] | Image+Text | | 6,000 | Acc |
| | Visual Grounding | RefCOCO [13] | Image+Text | 4 | 6,000 | IoU |
| | Caption Generation | RefCOCO | Image+Text | | 6,000 | CIDER/ROUGE_L |
| | CLS | COCO [18] | Image | | 6,000 | Acc |
| Fed-Med | VQA | SLAKE & VQA-RAD [15] | Image+Text | | 3,846 | Acc |
| | Detection | RadGenome-Chest CT | Image+Text | 3 | 8,744 | IoU |
| | Report Generation | MIMIC-CXR [12] | Image+Text | | 8,000 | CIDER/ROUGE_L |

reduction in VLM generalization by using dynamic adapter designs and a globally shared semantic space. FedMLLM [35] introduces a benchmark for evaluating federated fine-tuning performance of MLLMs across heterogeneous scenarios. However, these approaches do not systematically explore critical issues such as vision language model architecture, the interplay of different modules, and the intricacies of multi-task collaborative training within the FL context.

## 3 Federated Vision-Language Benchmark Datasets

Current federated benchmarks [35] exhibit two fundamental limitations in task coverage. First, while claiming multimodal capabilities, existing works predominantly focus on only two basic task types—VQA and classification—while ignoring more complex but critically important multimodal tasks such as report generation and visual localization. Second, and more importantly, there exists a complete absence of datasets supporting federated multi-modal multi-task learning scenarios, despite their practical significance in real-world applications where different clients may need to handle distinct multimodal tasks. To bridge the gaps, we develop six novel federated datasets through two synergistic efforts On the single-task front, we construct four specialized benchmarks (Fed-FGVC, Fed-SLAKE, Fed-ScienceCap, and Fed-RadGenome) that significantly expand beyond conventional VQA and classification to include caption generation and visual localization tasks, with careful consideration of both IID and non-IID data distributions. More innovatively, we pioneer two multi-task federated datasets (Fed-Nature and Fed-Med) that for the first time enable collaborative instruction tuning across interconnected multi-task and multimodal objectives, filling a crucial void in current FL research infrastructure.

**Fed-FGVC: A Classification Vision-Language FL Dataset.** FGVC-Aircraft [22] is a dataset designed for fine-grained visual classification of aircraft. Based on the key attribute "manufacturer"(30 categories), we distribute the data among up to 30 clients, ensuring that every three categories are evenly distributed or merged, resulting in IID and non-IID partitions. Additionally, four heterogeneous partitions are generated using varying Dirichlet coefficients, resulting in a Fed-FGVC dataset with six partitions to benchmark multimodal language models on fine-grained image understanding.

**Fed-ScienceCap: A Caption Generation Vision-Language FL Dataset.** ScienceQA [21] is a comprehensive dataset encompassing various question types from real science exams across different disciplines. We screened image-description pairs and excluded categories with fewer than 100 samples by "category". The remaining 27 categories were evenly distributed or merged to a maximum of 27 clients to create IID and non-IID partitions. The resulting Fed-ScienceCap dataset provides two partitioning schemes to evaluate models on image semantic understanding in natural sciences.

**Fed-SLAKE: A Visual Question Answering Vision-Language FL Dataset.** SLAKE [19] is a dataset for medical vision problems, covering various modalities, organs, and both closed and open questions. We first excluded question types with fewer than 20 samples and then used uniform and complete partitioning by "modality" to create IID and non-IID partitions among 3 clients.

**Fed-RadGenome: A Visual Detection Vision-Language FL Dataset.** RadGenome-Chest CT [42] is a multimodal dataset containing segmentation masks and region-specific reports for 3D chest CT scans. We extracted two 2D cross-sectional images from each 3D volume, along with masks for three organs (heart, lung, and abdomen) and their corresponding reports. Using uniform and complete

category division methods, we distributed the data among 3 clients, resulting in the Fed-RadGenome dataset, which includes over 8,000 samples and both IID and non-IID partitioning methods.

**Fed-Nature: A Natural Multitask Vision-Language FL Dataset.** Fed-Nature integrates three public vision-language datasets — COCO [18] (classification), RefCOCO [13] (visual grounding and captioning generation), and COCO-QA [29] (VQA) — by linking their cross-modal annotations through shared image IDs. We map each specific task to a dedicated client, creating four clients that jointly support VQA, classification, visual grounding, and caption generation tasks.

**Fed-Med: A Medical Multitask Vision-Language FL Dataset.** Fed-Med unifies chest-related medical question answering, detection, report generation, and various other data sourced from the SLAKE [19] (VQA), MIMIC-CXR [12] (report generation), VQA-RAD (VQA) [15], and RadGenome-Chest CT [42] (detection) datasets. Similar to Fed-Nature, we map each specific task to a client, creating three clients that jointly support VQA, report generation, and detection.

More details about the datasets and their partitions are provided in the supplementary file.

# 4   FedVLMBench Framework

To make our FedVLMBench framework compatible with standard FL protocols, it follows the same training process as conventional FL (e.g., FedAvg [23]), which involves a central server and $K$ clients. Each client holds a private multimodal dataset $D_k = \{(I^{(i)}, T^{(i)}, Res^{(i)}) \mid i = 1, 2, \ldots, N_k\}$ that includes images $I$, text $T$, and corresponding responses $Res$. The underlying optimization goal of our FedVLMBench can be formalized as follows:

$$arg \min_{w^s \in \mathbb{R}^d} \frac{1}{K} \sum_{k=1}^{K} \mathcal{L}_{\text{VLM}}^{(k)}(w_k), \tag{1}$$

where $\mathcal{L}_{\text{VLM}}^{(k)}(w_k)$ denotes the local loss function of client $k$, $N_k$ represents the number of samples in client $k$'s private dataset, $w_k$ represents the entire model parameters of client $k$, and $w^s$ denotes the trainable parameters.

Our FedVLMBench framework, as illustrated in Fig. 1, involves two mainstream VLM architectures: encoder-based and encoder-free. The former utilizes a connector $\mathcal{C}(\cdot; \theta_c)$ to map features extracted from the image encoder $\mathcal{E}$ into tokens, while the encoder-free approach directly employs the image tokenizer $\mathcal{T}_{\text{img}}$ to generate tokens. Both models use the text tokenizer $\mathcal{T}_{\text{text}}$ to encode textual information. For the encoder-based VLM, we employ four fine-tuning strategies that explore different orders and combinations of fine-tuning the connectors and LLMs. Specifically, the first strategy focuses on fine-tuning only the connector. The second strategy involves fine-tuning only the LLM using LoRA [10]. The third strategy entails simultaneously fine-tuning both the connector and the LLM with LoRA. Finally, the fourth strategy consists of fine-tuning the connector first, followed by the LLM using LoRA. For the encoder-free VLM, we only utilize LoRA to fine-tune the LLM.

In each FL communication round, the server first broadcasts the trainable parameters to each client. Then, clients conduct local fine-tuning and share the updated weights with the server for aggregation. The server aggregates these updates to update the global model and then re-broadcasts the trainable parameters to each client for the next round of fine-tuning. We will elaborate on this workflow in the following.

**Local Fine-Tuning Procedure.** For each round of local fine-tuning, we first update the trainable parameters with the received parameters, which may be partial due to varying training strategies. Then we perform stochastic gradient descent steps to update the trainable parameters. The update process is shown below:

$$w_k^s \leftarrow w_k^s - \eta_g \nabla_{w_k} \mathcal{L}_{\text{VLM}}^{(k)}(w_k), \tag{2}$$

where $w_k^s$ represents the trainable parameters of client $k$. For the encoder-based VLM, its composition varies according to the different fine-tuning strategies:

$$w_k^s = \begin{cases} \theta_c, & \text{fine-tune only the connector,} \\ \theta_{\text{LLM}}, & \text{fine-tune only the LLM using LoRA,} \\ \{\theta_c, \theta_{\text{LLM}}\}, & \text{fine-tune both the connector and LLM with LoRA simultaneously,} \\ \{\theta_c, \theta_{\text{LLM}}\}, & \text{fine-tune the connector and LLM with LoRA in order,} \end{cases} \quad (3)$$

where $\theta_c$ and $\theta_{LLM}$ represent the trainable parameters of the connector and LoRA in LLM, respectively. For the encoder-free VLM, we utilize LoRA to only fine-tune the parameters of the LLM, thus $w_k^s = \theta_{LLM}$.

**Global Aggregation.** Similar to common FL algorithms, the server performs weighted averaging of the trainable parameters as:

$$\bar{w}^s = \sum_{k=1}^{K} \alpha_k w_k^s, \quad (4)$$

where $\alpha_k$ is the aggregation weight for client $k$. In FedAvg [23], this weight is typically determined by the number of samples at the client, i.e., $\alpha_k = \frac{N_k}{\sum_{k=1}^{K} N_k}$.

## 5 Experiments

We systematically investigate federated fine-tuning VLM learning through three progressive dimensions. First, we explore how to efficiently fine-tune encoder-based VLMs within FL environments. We assess the impact of different connector layers (linear, 2-layer MLP, and 6-layer MLP), alongside various fine-tuning strategies under varying data distributions (IID/non-IID) to determine their influence on model performance. Next, we extend this analysis to compare encoder-based and encoder-free VLMs, revealing architectural disparities in handling data heterogeneity and task-specific sensitivities in single-task FL. Finally, leveraging these single-task FL findings, we evaluate federated multitask learning under both encoder-free and encoder-based VLM.

### 5.1 Experimental Setup

**Implement Details.**

For encoder-based VLM, we adopt LLaVA 1.5's architecture, utilizing a pre-trained CLIP visual encoder (ViT-B/32 [3, 26]) for visual feature extraction and LLAMA3.2-3B [24] as the language model. We investigate three connector layer configurations between visual and language modules: linear layer, 2-layer MLP, and 6-layer MLP. For encoder-free VLMs, we initialize Show-O [32] with its original pre-trained parameters for instruction fine-tuning. Across both architectures, we employ LoRA with rank 8 and scaling factor $\alpha$=32 for parameter-efficient tuning of LLM components. Additional implementation details are provided in the supplementary material.

Table 3: Performance comparison of connector layer types (linear layer, 2-layer MLP (Mlp2x), and 6-layer MLP (Mlp6x)) on FL fine-tuning on encoder-based VLM under ing IID data portions of Fed-SLAKE and Fed-ScienceCap datasets. F-C denotes the connector fine-tuning model, F-L denotes the LLM tuning model. LC denotes joint one-stage connector-LLM tuning and 2stage denotes the sequential fine-tuning of the connector and LLM. The best result is indicated in **bold**, while the second-best result is shown with underline. This performance notation scheme is consistent throughout the paper unless explicitly stated otherwise.

| Mode | Method | Fed-SLAKE | | | Fed-ScienceCap | | |
|------|--------|--------|-------|-------|--------|-------|-------|
| | | Linear | Mlp2x | Mlp6x | Linear | Mlp2x | Mlp6x |
| F-C | Central | 0.799 | 0.788 | 0.734 | 7.239/0.879 | 7.361/0.889 | 7.274/0.881 |
| | FedAvg | 0.726 | 0.783 | 0.759 | 7.069/0.867 | 7.283/0.882 | 6.991/0.866 |
| F-L | Central | **0.837** | 0.834 | 0.531 | **7.534**/0.898 | 7.459/0.896 | 5.784/0.833 |
| | FedAvg | 0.787 | 0.806 | 0.794 | 7.498/0.893 | 7.338/0.889 | 5.727/0.832 |
| F-CL | Central | 0.824 | **0.843** | 0.739 | 7.521/**0.899** | **7.550/0.901** | 7.366/0.892 |
| | FedAvg | 0.819 | 0.823 | 0.802 | 7.468/0.896 | 7.521/0.899 | 7.274/0.886 |
| F-2stage | Central | 0.815 | 0.830 | **0.817** | 7.424/0.892 | 7.414/0.894 | **7.491/0.894** |
| | FedAvg | 0.808 | 0.811 | 0.797 | 7.216/0.878 | 7.290/0.883 | 7.226/0.883 |

**Baseline FL Algorithms.** We evaluate five representative FL approaches spanning classical and adaptive heterogeneity optimization paradigms: FedAvg [23], FedProx [16], FedAvgM [9], FedYogi [28] and FedAdam [28]. To establish performance ceilings, we include a Central baseline trained on aggregated client data. More implementation details are provided in the supplementary material.

## 5.2 How to Efficiently Fine-tune Encoder-based VLM in FL?

Our initial exploration focuses on assessing the impact of various popularly utilized connection layers (linear, 2-layer MLP, and 6-layer MLP) along with different fine-tuning strategies on the performance of encoder-based VLM in FL.

**Which connector type—linear, 2-layer MLP, or 6-layer MLP—is most effective for FL fine-tuning of encoder-based VLMs?** As shown in Tab. 3, both the simple linear layer and the 2-layer MLP demonstrate superior performance across a range of fine-tuning strategies and tasks. In contrast, the more complex 6-layer MLP connector results in a significant reduction in performance in both the FL and Central settings, despite an increase in model parameters. This suggests that the added complexity in the connector does not necessarily translate to better performance in FL. The performance of linear layer in FL, while appearing effective and simple, is derived from optimal hyperparameter tuning, including the selection of the most favorable random seeds. In practice, linear layer is highly susceptible to parameter initialization (i.e., random seeds) in FL, resulting in significant fluctuations in training outcomes (see figure in the supplement). This sensitivity is particularly pronounced when each client has limited data—a common scenario in FL applications (see supplementary file for results). Based on these findings, we conclude that:

> **Takeaway 1**: Compared to a simple linear layer and a complex 6-layer MLP, a 2-layer MLP emerges as the most effective connector regarding performance, computational efficiency, and training stability for fine-tuning VLMs in FL.

Based on previous experimental findings, we employ a 2-layer MLP as the connection layer for all subsequent experiments in this study.

**How should we select FL fine-tuning strategies for different tasks in encoder-based VLMs?** In the context of federated fine-tuning in encoder-based VLMs, a key question arises: Which fine-tuning strategy is most effective: (1) connector-only (C, denoted as F-C), (2) LLM-only (L, denoted as F-L), (3) joint connector-LLM tuning (CL, denoted as denoted as F-CL), or (4) two-stage sequential tuning (C→L, denoted as F-2stage). We systematically evaluate these approaches across diverse vision-language tasks under FL constraints.

We begin by examining the impact of fine-tuning either the connector or the LLM across different tasks in FL set-

Table 4: Quantitative comparison of four fine-tuning strategies on multi-type task datasets with IID and non-IID distributions.

| Mode | Method | Fed-SLAKE | | Fed-ScienceCap | | Fed-FGVC | |
|------|--------|-----------|---------|----------------|---------|----------|---------|
| | | IID | Non-IID | IID | Non-IID | IID | Non-IID |
| F-C | FedAvg | 0.783 | 0.775 | 7.285/0.882 | 7.249/0.881 | 0.724 | 0.585 |
| | FedProx | 0.734 | 0.750 | 7.293/0.885 | 7.250/0.881 | 0.726 | 0.586 |
| | FedAdam | 0.741 | 0.735 | 7.127/0.876 | 7.137/0.876 | 0.694 | 0.522 |
| | FedAvgM | 0.754 | 0.747 | 7.252/0.880 | 7.238/0.881 | 0.696 | 0.510 |
| | FedYogi | 0.745 | 0.736 | 7.125/0.877 | 7.104/0.874 | 0.695 | 0.511 |
| F-L | FedAvg | 0.806 | 0.802 | 7.355/0.890 | 7.342/0.889 | 0.647 | 0.529 |
| | FedProx | 0.800 | 0.780 | 7.331/0.889 | 7.311/0.887 | 0.637 | 0.488 |
| | FedAdam | 0.783 | 0.771 | 7.194/0.885 | 7.125/0.881 | 0.627 | 0.460 |
| | FedAvgM | 0.789 | 0.786 | 7.287/0.890 | 7.305/0.890 | 0.602 | 0.469 |
| | FedYogi | 0.782 | 0.769 | 7.153/0.884 | 7.123/0.881 | 0.623 | 0.467 |
| F-CL | FedAvg | **0.823** | **0.827** | **7.501/0.898** | **7.476/0.897** | 0.721 | 0.603 |
| | FedProx | 0.816 | 0.796 | 7.500/**0.898** | 7.440/**0.897** | 0.718 | 0.548 |
| | FedAdam | 0.777 | 0.774 | 7.282/0.891 | 7.319/0.891 | 0.671 | 0.528 |
| | FedAvgM | 0.784 | 0.768 | 7.359/0.893 | 7.351/0.892 | 0.677 | 0.514 |
| | FedYogi | 0.783 | 0.774 | 7.277/0.890 | 7.287/0.890 | 0.675 | 0.511 |
| F-2stage | FedAvg | 0.811 | 0.814 | 7.334/0.884 | 7.281/0.883 | **0.730** | **0.614** |
| | FedProx | 0.773 | 0.785 | 7.262/0.883 | 7.221/0.880 | 0.715 | 0.591 |
| | FedAdam | 0.782 | 0.777 | 7.315/0.887 | 7.315/0.887 | 0.713 | 0.539 |
| | FedAvgM | 0.793 | 0.794 | 7.369/0.889 | 7.380/0.889 | 0.708 | 0.565 |
| | FedYogi | 0.785 | 0.782 | 7.310/0.886 | 7.310/0.886 | 0.717 | 0.561 |

tings. As detailed in Tab. 4, for the text-dominant tasks (e.g. the VQA on Fed-SLAKE and caption generation on Fed-ScienceCap datasets), LLM tuning (F-L) significantly outperforms connector-only tuning (F-C), and yields results comparable to full-model tuning (F-CL and F-2stage). Conversely, for vision-focused tasks (e.g., fine-grained image classification tasks on Fed-FGVC), connector tuning (F-C) achieves results comparable to full-model tuning (F-CL and F-2stage) while substantially outperforming LLM-only adaptation (F-L). This suggests that text-driven tasks benefit from updating linguistic knowledge, whereas vision-centric tasks require refined visual-textual alignment.

> **Takeaway 2**: In federated fine-tuning of VLMs, prioritizing LLM fine-tuning enhances performance in text-centric tasks, such as VQA and caption generation, while fine-tuning the connector is more effective for visually-driven tasks like image classification.

Subsequently, we compare full-model fine-tune strategies (F-CL vs. F-2stage). In traditional VLM fine-tuning, it is commonly believed that tuning the connector before the LLM is preferred. However,

Table 5: Performance comparison of different VLM architectures on various single-task datasets with IID and non-IID distributions.

| Mode | Method | Fed-SLAKE | | Fed-ScienceCap | | Fed-FGVC | | Fed-RadGnome | |
|---|---|---|---|---|---|---|---|---|---|
| | | IID | Non-IID | IID | Non-IID | IID | Non-IID | IID | Non-IID |
| **Encoder-based** | Central | 0.843 | | 7.550/0.901 | | 0.764 | | 0.584 | |
| | FedAvg | **0.823** | **0.827** | **7.501**/0.898 | **7.476**/0.897 | 0.721 | **0.603** | 0.565 | 0.484 |
| | FedProx | 0.816 | 0.796 | 7.500/0.898 | 7.440/0.897 | 0.718 | 0.548 | 0.535 | 0.462 |
| | FedAdam | 0.777 | 0.774 | 7.282/0.891 | 7.319/0.891 | 0.671 | 0.528 | 0.550 | 0.529 |
| | FedAvgM | 0.784 | 0.768 | 7.359/0.893 | 7.351/0.892 | 0.677 | 0.514 | 0.542 | 0.511 |
| | FedYogi | 0.783 | 0.775 | 7.277/0.890 | 7.287/0.890 | 0.675 | 0.511 | 0.556 | **0.536** |
| **Encoder-free** | Central | 0.784 | | 7.462/0.899 | | 0.739 | | 0.580 | |
| | FedAvg | 0.777 | 0.761 | 7.470/**0.902** | 7.421/**0.899** | 0.721 | 0.493 | **0.604** | 0.485 |
| | FedProx | 0.769 | 0.734 | 7.456/0.901 | 7.363/0.897 | 0.679 | 0.440 | 0.565 | 0.460 |
| | FedAdam | 0.747 | 0.732 | 7.241/0.894 | 6.850/0.881 | 0.689 | 0.471 | 0.597 | 0.472 |
| | FedAvgM | 0.776 | 0.743 | 7.398/0.899 | 7.402/**0.899** | **0.723** | 0.453 | 0.596 | 0.435 |
| | FedYogi | 0.749 | 0.737 | 7.221/0.893 | 7.267/0.894 | 0.686 | 0.467 | 0.599 | 0.461 |

our findings present an intriguing contrast. As illustrated in Table 4, fine-tuning both the connector and the LLM simultaneously (strategy F-CL) often results in superior or comparable outcomes compared to the sequential two-stage approach (strategy F-2stage), while also reducing computational overhead.

> **Takeaway 3**: For encoder-based VLMs in FL environments, concurrent fine-tuning of both the connector and the LLM outperforms sequential training connector first and then LLM in FL, balancing performance gains with computational efficiency.

Based on these experimental findings, we adopt the F-CL as the federated tuning strategy for all subsequent experiments in this study.

**What's the impact of data heterogeneity on federated fine-tuning of encoder-based VLMs?**
Building upon our analysis of FedAvg under IID settings, we now investigate how data heterogeneity affects different VLM tasks by establishing both IID and non-IID distributions across different tasks. As shown in Tab. 4, for text-centric tasks (such as visual question answering and caption generation), there is no significant difference in performance among the various fine-tuning methods under IID and non-IID conditions. However, vision-dependent tasks (Fed-FGVC) exhibit a significant performance drop of approximately 20% under non-IID settings compared to IID baselines. Notably, traditional FL optimizers like FedProx and FedYogi fail to address this performance degradation. This conclusion is further reinforced by experiments on non-IID datasets generated via Dirichlet distributions with varying heterogeneity levels, as demonstrated in figure in the supplement. These findings highlight the need for new approaches specifically designed to handle the unique challenges of federated fine-tuning for encoder-based VLMs, particularly for vision-centric tasks under non-IID conditions.

> **Takeaway 4**: Encoder-based VLMs maintain robustness on text-centric federated tasks under data heterogeneity, but exhibit significant performance drops for vision-centric tasks under non-IID conditions. Current FL optimization methods show limited effectiveness, calling for novel solutions tailored for vision-dominant multimodal FL learning.

### 5.3  How Do Different VLM Architectures Respond to Data Heterogeneity in FL?

Building on our analysis of encoder-based VLMs (Sec. 5.2), we systematically compare encoder-free architectures under identical FL conditions (IID/non-IID data, multitask scenarios). Unlike encoder-based models that separate visual and linguistic components with trainable connectors, encoder-free VLMs operate as unified frameworks without explicit alignment modules (connectors). As shown in Tab.5, encoder-free VLMs exhibit no significant performance variation on text-centric tasks (Fed-SLAKE and Fed-ScienceCAP) between IID and non-IID conditions, mirroring the behavior of encoder-based VLMs. This suggests that text-driven tasks inherently benefit from the linguistic priors of LLMs, regardless of architectural differences. For vision-dependent tasks (Fed-FGVC classification and Fed-RadGenome detection), both architectures suffer performance degradation under non-IID data. However, the performance drop for the encoder-free model on non-IID data is more pronounced than that of the encoder-based model on the vision-centric Fed-FGVC and Fed-RadGenome datasets. This disparity is likely due to the absence of trainable connectors, suggesting that learnable connectors can mitigate some challenges associated with data heterogeneity. Furthermore, consistent with our

Table 6: Quantitative comparison on Fed-Nature and Fed-Med datasets. MT-Central refers to centralized training on the centralized multi-task dataset.

| Mode | Method | Fed-Nature | | | | Fed-Med | | |
|---|---|---|---|---|---|---|---|---|
| | | VQA Acc↑ | Caption Generation CIDER↑ ROUGE_L ↑ | Visual Grounding IoU ↑ | Classification Acc↑ | VQA Acc↑ | Report Generation CIDER↑ ROUGE_L ↑ | Detection IoU↑ |
| **Encoder-based** | MT-Central | 0.755 | 0.872/0.358 | 0.405 | **0.913** | 0.674 | 2.101/0.595 | **0.616** |
| | FedAvg | 0.756 | 0.794/0.336 | 0.357 | 0.911 | **0.698** | **2.132/0.599** | 0.588 |
| | FedProx | 0.711 | 0.807/0.350 | 0.352 | 0.893 | 0.667 | 1.929/0.574 | 0.615 |
| | FedAdam | 0.742 | 0.810/0.344 | 0.386 | 0.901 | 0.683 | 2.054/0.589 | 0.567 |
| | FedAvgM | 0.735 | 0.788/0.336 | 0.393 | 0.912 | 0.664 | 1.921/0.576 | 0.592 |
| | FedYogi | 0.744 | 0.784/0.341 | 0.395 | 0.900 | 0.682 | 1.986/0.583 | 0.588 |
| **Encoder-free** | MT-Central | 0.752 | 0.912/0.361 | **0.465** | 0.874 | 0.610 | 1.922/0.575 | 0.581 |
| | FedAvg | **0.781** | 0.930/0.363 | 0.449 | 0.888 | 0.607 | 1.887/0.566 | 0.578 |
| | FedProx | 0.610 | 0.938/0.376 | 0.404 | 0.786 | 0.584 | 1.515/0.538 | 0.532 |
| | FedAdam | 0.739 | **1.090/0.402** | 0.460 | 0.885 | 0.651 | 1.806/0.564 | 0.579 |
| | FedAvgM | 0.761 | 1.010/0.390 | 0.426 | 0.886 | 0.634 | 1.790/0.555 | 0.604 |
| | FedYogi | 0.742 | 1.072/0.398 | 0.456 | 0.893 | 0.654 | 1.819/0.564 | 0.577 |

earlier findings in Sec. 5.2, traditional FL optimizers (e.g., FedProx, FedYogi) demonstrate limited efficacy in mitigating performance degradation for both architectures under non-IID conditions. This emphasizes the need for architecture-aware FL optimization strategies specifically tailored to address heterogeneity challenges in vision-centric VLM tasks.

> **Takeaway 5**: Both encoder-based and encoder-free VLMs exhibit robust performance on text-centric tasks under non-IID conditions, while vision-centric tasks show pronounced sensitivity to non-IID, with encoder-free VLMs exhibiting larger performance drops. Current FL optimization methods show limited effectiveness in both encoder-free and encoder-based VLMs, calling for novel solutions addressing vision-centric heterogeneity challenges.

### 5.4 How Do Various FL VLM Architectures Perform in Real-world FL Multi-task Scenarios?

Here, we investigate various VLM architectures and FL algorithms on the two multi-task FL datasets (Fed-Nature and Fed-Med). Our evaluation on real-world non-IID multitask FL benchmarks reveals a striking divergence from single-task FL observations: while single-task FL struggles with vision-centric performance degradation under non-IID data, federated multitask training achieves near-ceiling performance comparable to centralized training across both text- and vision-centric tasks, regardless of VLM architectures, see Tab.6. Additionally, while there is no clear winner among the existing FL algorithms on multi-task learning, the naive FedAvg provides more stable performance across various tasks compared to other FL-optimized methods. These findings underscore the viability of FL multitask learning as a privacy-preserving alternative to centralized training in real-world multi-task vision-language systems, particularly given the growing prevalence of multitask VLM deployments.

> **Takeaway 6**: Both encoder-based and encoder-free VLMs achieve near-ceiling centralized performance in real-world federated multitask learning, demonstrating their viability as privacy-preserving alternatives in multitask VLM deployments.

## 6 Conclusion

We present FedVLMBench, the first comprehensive benchmark for federated VLM fine-tuning, addressing critical gaps in architectural diversity (encoder-based vs. encoder-free VLMs), task coverage, and multi-task FL scenarios. Through systematic evaluation across 6 datasets, 5 FL algorithms, and 4 fine-tuning strategies, we demonstrate that 2-layer MLP connectors with concurrent connector-LLM tuning optimize encoder-based VLM performance, identify task-specific tuning strategies (LLM tuning for text-centric vs. connector-tuning for vision-centric tasks), and reveal that multi-task FL achieves near-centralized accuracy despite non-IID data. Notably, our findings reveal that conventional FL optimization methods for vision-centric tasks (e.g., detection) exhibit higher sensitivity to data heterogeneity than text-centric tasks in federated VLM tuning, demanding novel solutions addressing vision-centric heterogeneity challenges. We hope this work provides foundational support for advancing federated VL systems in real-world applications where data decentralization and task diversity coexist.

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
