# FedVLMBench: Benchmarking Federated Fine-Tuning of Vision-Language Models
## --Supplementary Materials--

## A   Additional Information about the Dataset

### A.1   Distribution of Dataset Attributes.

As shown in Fig.1, we present the category distribution for four data sources: Fed-FGVC, Fed-ScienceCap, Fed-SLAKE, and Fed-RadGenome. Each dataset contains a diverse array of data points, each with multiple attributes. This richness in diversity makes them ideal for constructing a comprehensive FL dataset.

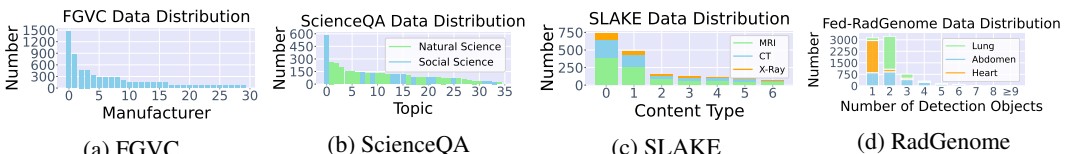

| (a) FGVC | (b) ScienceQA | (c) SLAKE | (d) RadGenome |

Figure 1: Data distribution of the original datasets. We divide them into clients with different degrees of data heterogeneity based on their labels, such as "Real World Non-IID Distribution" which strictly adheres to the established label categories.

As illustrated in Fig. 2, we also present the t-SNE visualization of text and image embeddings from different tasks in Fed-Nature and Fed-Med datasets. The images and texts of the two datasets exhibit certain similarities, indicating an overlap in knowledge among different clients. This overlap is conducive to collaborative training, which can improve performance in various tasks.

## B   More Implement Details

### B.1   Training Details

All experiments are conducted on NVIDIA L40S GPUs. The learning rate is initialized to 1e-4 and optimized using a cosine annealing scheduler to balance convergence speed and stability. The batch size and maximum sequence length for the model architecture are set to 8 and 512, respectively. For the encoder-free model, the codebook size for the image tokens is 8,192. To ensure fairness in evaluation, all additional hyperparameters are adjusted to their default values for benchmark comparison. Hardware-specific optimizations, including mixed precision training and gradient checkpointing, are uniformly applied across all runs to minimize resource discrepancies.

Submitted to 39th Conference on Neural Information Processing Systems (NeurIPS 2025). Do not distribute.

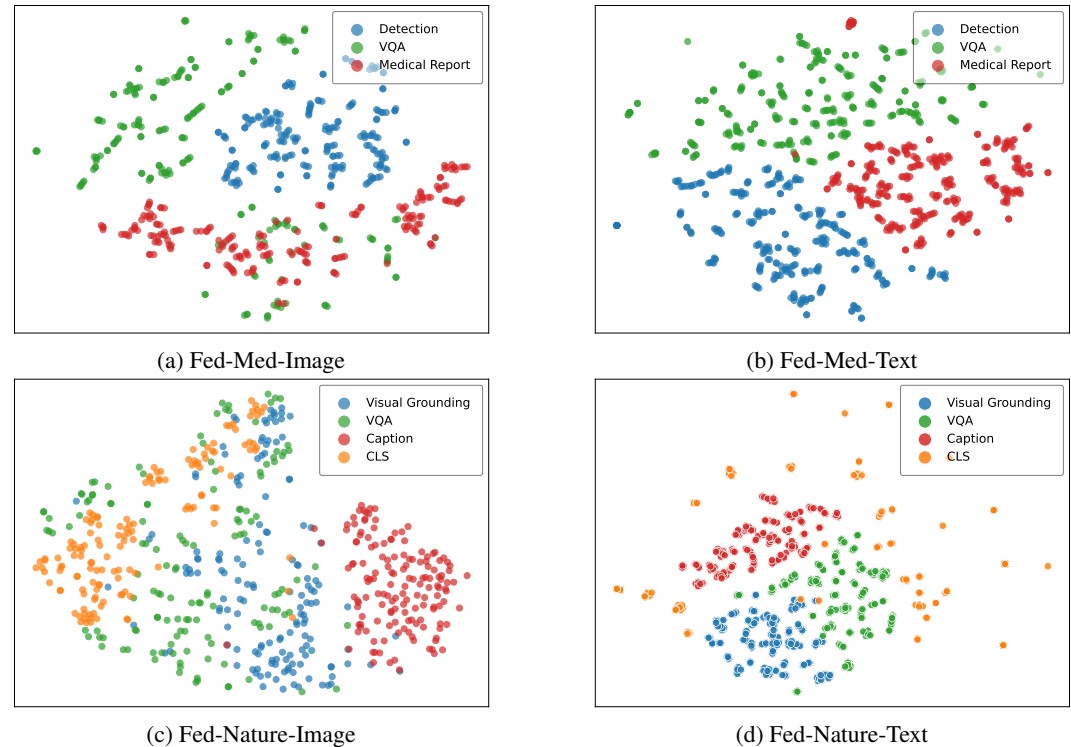

|  (a) Fed-Med-Image | (b) Fed-Med-Text |
| --- | --- |
| (c) Fed-Nature-Image | (d) Fed-Nature-Text |

Figure 2: The t-SNE visualization of text and image embeddings from different tasks in Fed-Med and Fed-Nature. Each color denotes one task. We can see clustering phenomenon of one client's data and that clients' data are diverse.

## B.2 Metrics

For the four types of downstream tasks discussed in our benchmark, we employ classic metrics to evaluate the performance of various methods.

1) **Image Classification** primarily employs Accuracy (Acc) to quantify the proportion of correctly predicted labels relative to the total number of samples.

2) **Text Generation** employs CIDEr (Consensus-based Image Description Evaluation), which weights n-gram similarity using TF-IDF to prioritize informative and diverse outputs. Additionally, ROUGE-L evaluates fluency through unigram recall and longest common subsequence alignment.

3) **Visual Question Answering (VQA)** also utilizes Task-Specific Accuracy (Acc), which evaluates exact matches or provides partial credit for semantically equivalent answers.

4) **Detection** relies on Mean Intersection over Union (mIoU), which measures the overlap between the predicted and ground-truth bounding boxes.

## B.3 Datasets

We provide details on the specific partitions for the following four datasets:

1) **Fed-FGVC:** Utilizing the key attribute "manufacturer" from the FGVC dataset, which consists of 30 categories, we create six distinct partitions.
   **IID Distribution**: We can allocate up to all 30 categories across a maximum of 30 clients, ensuring that each client receives a representative sample. This results in a balanced distribution among clients.
   **Real World Non-IID Distribution**: We can also distribute the 30 categories across a maximum of 30 clients by grouping the categories into different sets, assigning each group to a different

client. This approach creates imbalanced data distributions, reflecting real-world scenarios where clients have varying amounts of data.

**Simulated Non-IID Distribution**: We simulate the non-IID distribution with the Dirichlet Distribution. Specifically, we can similarly allocate the 30 categories across up to 30 clients, varying the alpha parameter to values of 0.01, 0.5, 5, and 100. Each setting generates data sets with different degrees of data heterogeneity, allowing us to analyze the impact of the data distribution on model performance.

In our experiments, we utilize a total of 5 clients across all distribution methods.

2) **Fed-ScienceCap:** We screen image-description pairs and exclude categories with fewer than 100 samples based on 'category'. We then distribute the remaining 27 image-description pairs among different clients, creating two distinct partitions.

**IID Distribution**: We can allocate up to 27 categories across a maximum of 27 clients, evenly distributing different categories to each client to ensure balanced representation.

**Real World Non-IID Distribution**: We can also group the 27 categories while maintaining a similar number of data points across the clients, reflecting varying data distributions.

In our experiments, we utilize a total of 5 clients for both distribution methods.

3) **Fed-SLAKE:** We first exclude question types with fewer than 20 samples, then use uniform and complete partitioning by "modality" to create IID and Non-IID partitions among 3 clients.

**IID Distribution**: We evenly distribute all data to three clients based on different medical imaging modalities (X-ray, CT, and MRI), resulting in a balanced partition.

**Real World Non-IID Distribution**: We allocate all images to the three clients according to modality, creating a partition that reflects varying data distributions.

4) **Fed-RadGenome:** We first extract 2D images from the 3D volume segmentation mask in the RadGenome-Chest CT dataset. Specifically, we extract two axial slices at normalized heights of 35% (z=0.35) and 50% (z=0.5) along the superior-inferior axis, generating two 2D images from the same case. We select three organs—lung, abdomen, and heart—as our detection targets, each potentially containing multiple detection targets. For each irregularly shaped detection target, we generate axis-aligned bounding boxes based on spatial extremal points.

**IID Distribution**: We evenly distribute all data to three clients based on organ type, resulting in a balanced IID partition that ensures that each client has a similar representation of the data.

**Real World Non-IID Distribution**: We allocate the data to the three clients according to specific organ categories, creating a Non-IID partition that reflects varying data distributions among clients.

## B.4 Baseline

We evaluate five representative FL approaches spanning classical and adaptive heterogeneity optimization paradigms: FedAvg [3], FedProx [2], FedAvgM [1], FedYogi [4] and FedAdam [4].

1) **FedAvg**: A foundational method in federated learning (FL) that performs local multi-round training on clients before uploading model parameters. It aggregates these parameters through weighted averaging, significantly reducing communication rounds while ensuring data privacy.

2) **FedProx**: This method builds upon FedAvg by introducing a proximal regularization term to limit the deviation of local models from the global model. The regularization strength is set to $\mu = 0.01$, which helps alleviate convergence issues caused by data and device heterogeneity, thereby enhancing model robustness in non-IID data distributions.

3) **FedAvgM**: Enhancing the parameter aggregation process of FedAvg, FedAvgM incorporates a momentum mechanism that introduces a momentum term during global model updates. This approach smooths out historical gradient directions, accelerating convergence and stabilizing updates, particularly in complex client data distributions. The momentum parameters are set as follows: $\beta_1 = 0.9$ for the first moment and $\beta_2 = 0.99$ for the second moment.

4) **FedYogi**: An adaptive FL algorithm based on the Yogi optimizer, which dynamically adjusts learning rates to effectively tackle non-convex optimization problems. This method estimates second-order moments of gradients, improving global update directions in heterogeneous data scenarios. The adaptation is controlled by a parameter $\tau = 0.001$, which influences the learning rate adjustments.

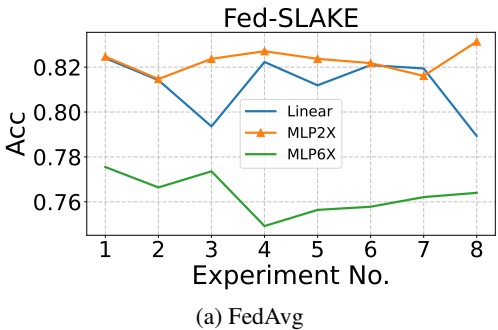
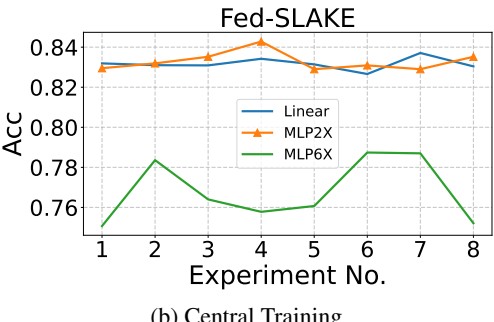

(a) FedAvg

(b) Central Training

Figure 3: Performance of different connection layers across multiple random experiments on the Fed-SLAKE dataset. We conduct eight evaluations using FedAvg and Central training respectively.(mentioned in Sec.5.2)

5) **FedAdam**: This method integrates the momentum mechanism and adaptive learning rates of the Adam optimizer into the federated framework. By leveraging weighted first and second moments of gradients, it achieves faster convergence and greater stability compared to FedAvg. The parameters include $\beta_1 = 0.9$ and $\beta_2 = 0.99$ for the momentum coefficients, facilitating effective global parameter updates.

# C   Additional Experimental Results

## C.1   Experiments on the Stability of Connector Training

As shown in Fig.3, we show the results of different connector types in two scenarios: FL and centralized training. It can be seen that the performance of the linear layer in FL fluctuates significantly more than that in centralized training, which indicates that the linear layer is unstable in the FL scenario with limited data (each client in the FL scenario only has a portion of the data in the centralized training mode).

## C.2   Experiments on Varying Degrees of Data Heterogeneity

Fig.4 illustrates the performance of different FL algorithms across various levels of data heterogeneity. Notably, traditional FL optimizers such as FedProx and FedYogi do not effectively mitigate this performance degradation.

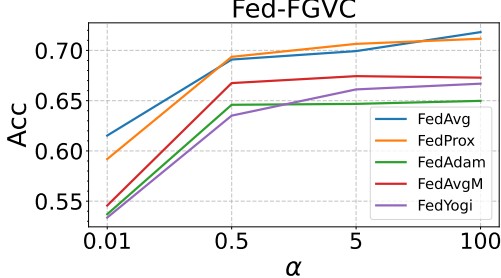

Figure 4: Performance of different FL algorithms at various levels of data heterogeneity on the Fed-FGVC dataset. $\alpha$ is the parameter of the Dirichlet distribution.(mentioned in Sec.5.2)