# OpenReview forum: "FedVLMBench: Benchmarking Federated Fine-Tuning of Vision-Language Models"
_NeurIPS.cc/2025/Datasets_and_Benchmarks_Track — Submitted to NeurIPS 2025 Datasets and Benchmarks Track_

### Official Review · Reviewer_ehBH · 2025-06-28

**Rating:** 4
**Confidence:** 3

**Summary:**

The authors propose FedVLMBench, the first systematic benchmark for federated fine-tuning of vision-language models (VLMs). This benchmark integrates two mainstream types of VLM architectures (encoder-based and non-encoder-based), four fine-tuning strategies (tuning only the connector, tuning only the LLM with LoRA, jointly tuning both the connector and the LLM, and sequentially tuning the connector followed by LoRA-based LLM tuning), five federated learning algorithms, and six multimodal datasets (Fed-FGVC, Fed-SLAKE, Fed-ScienceCap, Fed-RadGenome, Fed-Nature, and Fed-Med).

**Dataset Code Accessibility:**

Yes

**Dataset Code Comments:**

Both the datasets and code are fully provided, with complete content for all six datasets. The paper offers detailed instructions on environment setup, dataset download, and a clear file directory structure for the codebase. Configuration and usage guidelines for both encoder-based and encoder-free VLMs are also well documented. Additionally, the scripts for generating configuration files and running training are included, making the implementation easy to reproduce and use.

**Ethical Considerations:**

No, there are no or only very minor ethics concerns

**Final Justification:**

The final rating remains at 4, with the decision as borderline accept. The reasons are as follows: First, the paper itself has notable strengths, such as the motivation is clear and well-founded and the experiments are comprehensive. Second, the rebuttal provides meaningful zero-shot performance results and includes a discussion of the limitations. Although the responses on communication cost and different fine-tuning strategies only partially address my concerns, Tables A14 and A15 still offer insights into the comparison of communication costs and the impact of different fine-tuning combinations on performance. Taken together, I have decided to maintain my original rating.

**Limitations Weaknesses:**

1) For vision-language models (VLMs), in addition to the current centralized and federated results, including some zero-shot performance comparisons would provide more comprehensive insights;
2) When applying VLMs in federated learning, the communication cost can differ significantly from traditional FL settings. It would be beneficial to include a comparison and analysis of communication overhead;
3) On cross-task datasets such as Fed-Nature and Fed-Med, the impact of different fine-tuning strategies warrants further investigation;
4) The paper lacks a discussion of future directions, potential optimization strategies, and known limitations of the current approach.

**Strengths Contributions:**

1) The motivation is clear and well-founded, focusing on comparisons between encoder-based and non-encoder-based model architectures, as well as different fine-tuning strategies. In addition, the inclusion of tasks such as medical report generation and visual grounding, along with the construction of corresponding federated datasets, enhances support for multimodal and multi-task scenarios;
2) The experiments are comprehensive, covering six federated multimodal datasets and providing systematic analysis of different model architectures and fine-tuning strategies;
3) The paper is well-structured, logically organized, and clearly written.

---

> ### Author Rebuttal · Authors · 2025-07-31
>
> We sincerely thank the reviewer for the efforts and positive feedback. We are pleased that the motivation behind FedVLMBench is viewed as "clear" and "well-founded". Please see our response below regarding the specific comments.
>
> **Q1. For VLMs, in addition to the current centralized and federated Results, including some zero-shot performance comparisons would provide more comprehensive insight.**
>
> As suggested, we report the zero-shot performance of two VLMs (encoder-based and encoder-free) on four single-task and two cross-task datasets in Tab.A11, Tab.A12 and Tab.A13. Notably, despite providing sample responses, the pre-trained models performed poorly, primarily due to the challenges presented by the datasets. Fed-SLAKE involves medical open-ended question answering, where general pre-trained models typically lack relevant medical knowledge. Fed-FGVC focuses on fine-grained classification, and these models have limited ability to extract and distinguish fine-grained features. Additionally, Fed-RadGenome includes detection tasks with more complex response formats, further affecting performance. These results also indicate that the comparisons in our paper are fair and reliable, as the pre-trained models have not encountered the relevant datasets.
>
> Table A11：Zero-shot performance on four single-task datasets.
> |      Mode    | Fed-SLAKE| Fed-ScienceCap | Fed-FGVC | Fed-RadGnome_detection|
> |:-|:-:|:-:|:-:|:-:|
> | Encoder-base |0.088|0.306/0.327|0.041|0 |
> | Encoder-free |0.067|0.516/0.361|0.039|0 |
>
> Table A12：Zero-shot performance Fed-Nature datasets.
> | Mode         | VQA   | Caption Generation | Visual Grounding | Classification |
> |:-|:-:|:-:|:-:|:-:|
> | Encoder-base|0.086 | 0.228 / 0.239  | 0 | 0.075|
> | Encoder-free|0.101 | 0.422 / 0.245  | 0 | 0.015|
>
> Table A13：Zero-shot performance Fed-Med datasets.
> | Mode         | VQA   | Report Generation | Detection |
> |:-|:-:|:-:|:-:|
> | Encoder-base | 0.106 | 0.392 / 0.315 | 0 |
> | Encoder-free | 0.028 | 0.157 / 0.243 | 0 |
>
> **Q2. Concerns on comparison and analysis of communication overhead.**
>
> As suggested, we tested the communication cost of different FL methods on FED-SLAKE dataset for both IID and Non-IID distributions (see Tab.A14). The results indicate that, regardless of the VLM architecture, FedAvg converges consistently faster than other methods, which is consistent with our previous performance results reported in Tab.6 of the main paper. Furthermore, the encoder-based VLM demonstrates much faster convergence than the encoder-free VLM, which we attribute to the learnable connector’s ability to better align image and text tokens, facilitating more efficient optimization. These findings are consistent with Takeaway 5 in our main paper, where encoder-free VLMs exhibit larger performance drops compared to encoder-based VLMs.
>
> Table A14：The number of communication rounds required for each method to reach the predefined target performance (ACC:0.7) on the Fed-SLAKE dataset.
> | Method|Encoder-based| Encoder-based |Encoder-free|Encoder-free|
> |:-|:-:|:-:|:-:|:-:|
> | Method |IID | Non-IID |IID|Non-IID|
> | Fedavg|4| 8|8|15|
> | Fedprox|4|10|10|20|
> | FedAdam|14|15|29|40|
> | FedAvgM|11|21|25|32|
> | FedYogi|14|16|29|41|
>
>
> **Q3. The impact of various fine-tuning strategies on cross-task datasets.**
>
> As suggested, we reported the relevant experimental results in Tab.A15 for encoder-based VLM. The findings indicate that fine-tuning only the connector (F-C) significantly enhances performance in classification tasks, effectively aligning the image space with the feature space. Conversely, fine-tuning solely the LLM (F-L) proves beneficial for text-related tasks, such as Visual Question Answering (VQA) and Caption Generation. Notably, the performance of joint fine-tuning (F-CL) surpasses that of the two-stage fine-tuning (F-2stage). These conclusions align with **Takeaway 2** in Section 5.2. We will update the detailed findings in the revised version to reflect these insights more comprehensively.
>
> Table 15：Quantitative comparison of four fine-tuning strategies of encoder-based VLMs on Fed-Nature dataset.
> |   Mode   |  VQA  | Caption Generation| Visual Grounding | Classification|
> |:-|:-:|:-:|:-:|:-:|
> | F-C      | 0.593 |    0.534/0.269    |       0.2        |     0.869     |
> | F-L      | 0.69  |    0.677/0.326    |      0.321       |     0.892     |
> | F-CL     | 0.756 |    0.794/0.336    |      0.357       |     0.911     |
> | F-2stage | 0.677 |    0.586/0.289    |      0.24        |     0.89      |
>
> **Q4. The paper lacks a discussion of future directions, potential optimization strategies, and known limitations of the current approach.**
>
> Thank you for your valuable suggestion. We agree that it is important to discuss future directions, potential optimization strategies, and known limitations. In the revised version, we will add a dedicated section to address these aspects.
>
> Our current work focuses on fundamental challenges in federated learning for VLMs, such as data heterogeneity (non-IID distributions), diverse task types, and multi-task fine-tuning. However, we recognize that several important issues remain unaddressed. For example, as noted in our response to Q3, our analysis mainly concentrates on the final model performance, while a more detailed study of communication cost and efficiency is lacking. We believe that a comprehensive investigation into communication overhead in different FL settings would be a valuable direction for future research.
>
> Furthermore, our study is limited to horizontal federated learning. We did not explore other FL paradigms, such as vertical or hybrid FL, which are also highly relevant for VLMs and present interesting opportunities for future work. In addition, practical system-level challenges, including device capability heterogeneity, straggler clients, and communication bottlenecks, are not covered in our experiments, but are crucial for real-world FL applications. Another limitation is that we do not address privacy protection in the FL+VLM context. The intersection of privacy concerns and federated fine-tuning of large vision-language models could introduce new challenges and is worthy of further investigation.
>
> Finally, our experimental results reveal that, under non-IID conditions, there is notable performance degradation in vision-centric tasks such as detection and fine-grained visual classification. We also observe that federated multi-task fine-tuning still leaves room for improvement.
>
> We will make sure to include a comprehensive discussion of these limitations and possible future directions in our revision, and we hope these points will inspire further research in this area.

---

> > ### Comment · Reviewer_ehBH · 2025-08-01
> >
> > The authors have addressed some of my concerns; therefore, I will keep my score unchanged.

---

> > > ### Author Response · Authors · 2025-08-01
> > > **Thanks for Reply**
> > >
> > > Thank you for your prompt response. We are pleased to hear that some of your concerns have been addressed, and we appreciate the constructive comments that have contributed to improving our work. Your continued positive support is invaluable to us. If there are any remaining issues or additional questions, please do not hesitate to let us know—we would be happy to discuss them further.

---

### Official Review · Reviewer_gjMf · 2025-07-02

**Rating:** 5
**Confidence:** 5

**Summary:**

- This work focusing on comparing different VLMs finetuning settings in federated senarios.  Claims are at least 6 findings in federated setting, one of that is new to VLM literatures of one perspective, encoder-based and -free models, i.e., LLaVA (mainstream) and Show-o.

**Dataset Code Accessibility:**

Partly

**Dataset Code Comments:**

- The proposed data repo is public but with limited tutorial and guideline for building datasets and code repo.
- Repo is built on other repo, no mirror repo for reproducing risks.
- Some personal information in the repo is not cleared.
- No project reciept.

**Ethical Considerations:**

No, there are no or only very minor ethics concerns

**Final Justification:**

I've carefully read all reviewers' comments and the authors' rebuttal. Most of the main technical concerns are solved. Futher issues  may be fixed in the future revisoin but not current version. The related issues are as follows:
- LoRA's rank ablation.
- (Main, raise my rating if it's solved.) Rebuttal Q10 is biased. I mean that even if it's practical in real-world, the authors should provide references to support the claim. The two citations in Line 35 is not enough. These two works, released in Sep. and Nov., 2024, are cited only by this work and 2 other works proposed in 2025. How could this support as 'practical' and largely studied in literatures of VLM efficient finetuning in FL? (Duplicated citations in References 36 and 37)
- Citations and discussion to support practice of VLMs in real-world application in privacy-preserving cases, e.g., hospital. It's not about efficiency methods but VLMs.
- Please notice the different between data heterogeneity and client drift. Detailed dicussion and references are missed. Specificaaly, rebuttal Q8 from the authors.

**Limitations Weaknesses:**

- Limited models for each architecture are provided. Moreover, it's hard to compare in a benchmarking work, that focusing on different pretrained models with varied architecture. Works are not involved and discussed, of mainstream encoder-based models after LLaVA about freezing or fine-tuning the LLM part, e.g., VILA [1].
- Finetuning with only LoRA (rank of 8), is too low and limited for common VLM LoRA finetuning. It's not representative. Ranks of 32/64/128 and alpha of 256 are more common for models less than 10B, even in resource-constrained industrial application.
- It seems like the gap between full-parameter finetuning and LoRA finetuning is overlooked, benchmarking finetuning .
- Only knowledge-related, missing conventional VLM indicators such as hallucinations and domain-aware behaviors. Knowledge-related datasets are limited in science and medical domains. This make it far from a comprehensive benchmark for VLMs evaluation.
- Multi-task and multi-modal tasks are mixed-defined and not decoupled, e.g., In Line 126-158, medical and regular-context images are very different, but are categorized as a general vision modality, and the Nature dataset in turn contains classification, grounding, caption, and VQA. This is alongside the previously catogorized dataset, and this way of doing the artificial federal scenario segmentation processing by the original dataset can make subsequent use of bench metrics coupling without being able to break down the segmentation strengths of each model.
- The introduction is biased with logical and factual errors. Line 29-31 say that significantly improved zero-shot performance of VLM is enhanced by efficient finetuning methods, which is weird because efficient finetuning methods are trade-off between performance and efficient and degrade finetuned models' performance. Moreover, the most of the current popular VLMs involves vision tower (called encoder in this work), so that the motivation of involving encoder-free into benchmarking consideration should be explained in details.
- It's not clear whether data heterogeneity in VLMs with large amount of datasets is practical? Is it a vertical federated learning problem?
- The so-called practical applications, of VLMs finetuning in real world, are not evidentially supported. It seems like a compiled demand scenario, which is confusing, for example, VLM demanding in the hospital does not mean that all the data is private. The mentioned tasks, e.g., report generation, are not supported by any other works, tech. reports, or even in literature (Line 41-47).
- What kind of private data will need VLMs application. There is no even single support, resulting in this so-called practical motivation, is not practical at all in Line 37-48.

**Strengths Contributions:**

- Common senses in centralized training of VLMs are verified in the federated learning scenario set up in this paper, for example, that 2-layer MLP connector on VLMs' vision inputs is effective and efficient for both VLM feature alignment (i.e. VLM pretraining) and downstream finetuning (post-training, i.e., multi-task training).
- Training strategies of joint tuning for better zero-shot and in-context learning performance proposed in VILA [1] are verified in FL VLM finetuning cases that joint tuning make zero-shot performance better.
- Interesting novel names and categorization, about vision-centric and text-centric federated tasks in VLM, are related to new categorization of encoder-based and encoder-free VLMs, which is called adapter finetuning and early fusion in VLM finetuning literature.

[1] VILA: On Pre-training for Visual Language Models.

---

> ### Author Rebuttal · Authors · 2025-07-31
>
> We thank the reviewer for the time and effort in evaluating our work. Please see our response below regarding the specific comments.
>
> **Q1. Hard to compare in a benchmarking work on different pretrained models with varied architecture.**
>
> We would like to clarify that we ensure fair benchmarking by maintaining consistent experimental conditions within each VLM architecture. For instance, in Takeaway 2 of the main paper, we explore the impact of four fine-tuning strategies on different task types, using a 2-layer MLP connector for all encoder-based VLMs. All settings are kept consistent, ensuring that the comparisons within encoder-based VLMs are fair. In our subsequent experiments, we do not directly compare different architectures but instead focus on exploring performance drops within the same VLM structure across different settings. Therefore, the comparisons presented in our work are both fair and reasonable.
>
> **Q2. Limited models for each architecture. Discussion about mainstream encoder-based models after LLaVA.**
>
> As suggested, we have now incorporated support for the Qwen2.5-VL model, see Tab.A1 and Tab.A2 (Q1 to reviewer 3GTm).
> Will also add discussions with other encoder-based VLMs after LLaVA (e.g., VILA) in revision.
>
> **Q3. Finetuning with only LoRA (rank=8) is not representative.**
>
> Thank you for your valuable suggestion. We used the suggested LoRA parameters (rank=32, alpha=256) to verify the representative conclusions **Takeaway 1 5 6** based on encoder-based VLMs, following the same settings as in Tab.3, Tab.4, and Tab.5 of the main paper.
>
> First, we show the performance of different connectors based on Fed-SLAKE dataset for F-CL Mode (joint fine-tuning connector and LLM) in Tab.A8. It can be seen that 2-layer MLP is still the best choice. This is consistent with Takeaway 1 in our paper.
> Next, as shown in Tab.A9, we evaluate FedAvg's performance on the Fed-SLAKE and Fed-FGVC datasets. For the text-centric task (Fed-SLAKE), there is no significant difference in FedAvg's performance under IID and Non-IID conditions. However, for the vision-centric task (Fed-FGVC), its performance drop significantly under Non-IID conditions. This observation aligns with Takeaway 5 in our paper.
> Finally, as illustrated in Tab.A10, we present the performance difference between centralized training (MT-Central) and FedAvg in multi-task federated fine-tuning on Fed-Nature dataset. Overall, except for the detection task, FedAvg's performance in the other tasks is comparable to that of centralized training, which is consistent with Takeaway 6 in our paper.
>
> These results confirm that our main conclusions remain valid even with the revised LoRA parameters. We will include additional experiments in our revision.
>
> Table A8：Performance comparison of connector layer types on FL fine-tuning on encoder-based VLM undering IID data portions of Fed-SLAKE( LORA parameters: rank=32，alpha=256).
> |F-CL|Linear|Mlp2|Mlp6x|
> |:-|:-:|:-:|:-:|
> |Fedavg|0.792|0.803|0.761|
>
> Table A9：Experiment results on Fed-SLAKE and Fed-FGVC datasets.
> |Dataset| Fed-SLAKE|Fed-SLAKE|Fed-FGVC|Fed-FGVC|
> |:-|:-:|:-:|:-:|:-:|
> |Method|IID|Non-IID|IID|Non-IID|
> |Fedavg|0.803|0.801|0.728|0.622|
>
> Table A10：Experiment results on Fed-Nature dataset.
> |Mode|VQA|Caption Generation| Visual Grounding| Classification|
> |:-|:-:|:-:|:-:|:-:|
> |MT-Central|0.745|0.893/0.358|0.398|0.914|
> |Fedavg|0.736|0.813/0.343|0.339|0.909|
>
> **Q4. The gap between full-parameter finetuning and LoRA finetuning.**
> We would like to emphasize that our work is primarily focused on instruction tuning and parameter-efficient finetuning methods such as LoRA in the FL setting, as specifically outlined in the abstract (lines 2–3) and introduction (lines 27–30) of our paper. A comprehensive study of full-parameter finetuning in FL is beyond the scope of our current work.
>
> **Q5. Not a comprehensive benchmark for VLMs evaluation: absence of conventional VLM indicators (hallucinations & domain-aware behaviors) & knowledge-related datasets are limited in science & medical domains**
> We fully acknowledge that traditional VLM evaluation covers a broad range of metrics, including conventional task performance (e.g., accuracy), hallucination, and domain-aware behaviors analysis. However, our work is specifically focused on addressing the most critical gap in FL for VLMs: systematically benchmarking parameter-efficient fine-tuning methods in FL, focusing on foundational performance metrics such as classification accuracy and detection precision. The selected tasks—classification, segmentation, detection, captioning, and report generation—are among the most practical for VLM and FL applications. While hallucination and domain-aware behaviors analysis are valuable, we believe this should be explored after establishing robust baselines for core foundational performance.
>
> Our extensive FL datasets, settings, experiments, and results have also been highly recognized by other reviewers, including Reviewer 3GTm (“comprehensive”) & Reviewer YAmn (“thorough experiments”). Note that our benchmark not only covers the science and medical domains, but also includes basic nature domain tasks （Fed-FGVC & Fed-Nature).
>
> We hope the added explanations could clarify your concerns.
>
>
> **Q6. Multi-task and multi-modal tasks are mixed-defined and not decoupled**
>
> We respectfully disagree with the reviewer’s claim that “multi-task and multi-modal tasks are mixed-defined in our paper.” Below, we clarify the design choices underlying our construction of federated datasets and highlight how our benchmark addresses key limitations in the current literature.
>
> **Our primary motivation in constructing federated datasets is to overcome two fundamental limitations in existing FL multimodal benchmarks for VLMs. See below:**
>
> **1) Addressing limited task coverage in FL multimodal benchmarks**
> Most prior works on FL with multimodal datasets focus exclusively on two basic task types (VQA and classification), neglecting more complex yet practical tasks such as report generation and visual localization (see Tab.1 in our paper). To address this gap, we constructed four single-task FL multimodal datasets (Fed-FGVC, Fed-SLAKE, Fed-ScienceCap, and Fed-RadGenome). Importantly, each of these four FL datasets is dedicated to a single, well-defined task. In our experiments, we analyze FL performance independently for each task and setting (Tab.3–5), with no mixing of multi-task and multi-modal evaluations.
>
> **2) Filling the gap in federated multi-modal multi-task learning**
> A second critical limitation in existing research is the lack of datasets supporting federated multi-modal, multi-task learning—despite the prevalence and importance of such scenarios in practical applications. We introduce two pioneering multi-task federated datasets (Fed-Nature and Fed-Med).
> For these datasets, multi-task and multi-modal learning are intentionally combined and analyzed together by design, as this reflects real-world needs and use cases.
>
> **Q7. Bias in the introduction and justification for incorporating encoder-free VLMs.**
>
> Thank you for your careful reading and valuable feedback. We acknowledge the typographical error in Lines 29-31; we intend to convey that parameter finetuning methods can significantly enhance the generalization ability of VLMs. We will correct this in the revision.
>
> While most widely used VLMs incorporate vision towers, the need to separately train image encoders and LLMs increases system complexity. Recently, encoder-free VLMs have emerged as another major research direction in the field, following the development of encoder-based VLMs. Therefore, we believe that incorporating this type of model into our benchmark is essential for further exploration.
>
>
> **Q8. Whether data heterogeneity in VLMs with large amount of datasets is practical?**
>
> First, data heterogeneity is a highly practical and representative challenge for VLMs in FL, even when each client has access to large-scale datasets. In practical deployments, different organizations or clients naturally collect data from varying sources, under different conditions, and with diverse distributions. This leads to significant heterogeneity across local datasets, regardless of their size. Second, our proposed FL benchmarks are carefully designed to reflect a wide range of practical scenarios. Our FL datasets not only include cases where clients have large quantities of data, but also include cases where clients possess only a small or highly variable number of samples.
>
> We hope this addresses the practical relevance of data heterogeneity in federated VLMs.
>
>
> **Q9. A vertical FL problem?**
>
> Our work is based on the horizontal FL paradigm, not vertical FL. Will clarify this in revision.
>
> **Q10. VLMs finetuning is not practical in real world and are not evidence-supported.**
>
> We respectfully disagree with the claim that practical applications of VLM finetuning lack evidentiary support.
>
> First, the need for efficient adaptation of VLMs is well-established and driven by real-world resource constraints. The majority of institutions—such as universities, rural hospitals, and community clinics—simply do not have the computational resources or data scale to support full-model training. For these organizations, finetuning pre-trained VLMs is the only feasible way to leverage state-of-the-art models.
>
> Second, the need to handle sensitive, decentralized data in fields like medicine motivates combining FL and VLM finetuning. The reviewer’s skepticism about federated VLM applications overlooks thriving research frontiers. A simple Google Scholar search with "federated learning or collaborative learning or multi-center studies" AND ("report generation" OR "image segmentation" OR "image caption") yields thousands of papers, including papers in top conferences and journals, demonstrating active research in this area.
>
> **Q11. Data & code repo.**
>
> Updated as suggestions. See data & code repo for more details.

---

> ### Comment · Area_Chair_cq3X · 2025-08-05
>
> Gentle reminder. Please read through the authors' rebuttal and share any further comments. Thanks!

---

> ### Author Response · Authors · 2025-08-06
>
> We would like to kindly follow up to see if our previous response has adequately addressed all of your concerns. If there are any remaining issues or if you need further clarification, we would be more than happy to discuss with you and provide additional details and improvements. Your feedback is invaluable to us, and we truly appreciate your time and consideration.

---

> ### Comment · Reviewer_gjMf · 2025-08-07
>
> I appreciate the authors' efforts on rebuttal. The comments are posted and modified in the final justification. I raise my score from 2 to 3 borderline reject now, and may raise further if the main issue is solved. For your convenience, I put it below.
>
> I've carefully read all reviewers' comments and the authors' rebuttal. Most of the main technical concerns are solved. Futher issues  may be fixed in the future revisoin but not current version. The related issues are as follows:
> - LoRA's rank ablation.
> - **(Main, raise my rating if it's solved.)** Rebuttal Q10 is biased. I mean that even if it's practical in real-world, the authors should provide references to support the claim. The two citations in Line 35 is not enough. These two works, released in Sep. and Nov., 2024, are cited only by this work and 2 other works (an arxiv survey and FedNano) proposed in 2025. How could this support as 'practical' and 'widely studied' in literatures of VLM efficient finetuning in FL? (Duplicated citations in References 34 and 35)
> - Citations and discussion to support practice of VLMs in real-world application in privacy-preserving cases, e.g., hospital. It's not about efficiency methods but VLMs.
> - Please notice the different between data heterogeneity and client drift. Detailed dicussion and references are missed. Specificaaly, rebuttal Q8 from the authors.

---

> > ### Author Response · Authors · 2025-08-07
> > **Thanks for Reply**
> >
> > Thank you for your reply. We are pleased to hear that most of the main concerns have been addressed. Below, we clarify our responses to the remaining concerns:
> >
> > **Q12: "LoRA's rank ablation."**
> >
> > As mentioned in our response to Q3 and shown in Tables A8–A10 above,  we used the suggested LoRA parameters (rank=32, alpha=256) to verify the representative conclusions Takeaway 1 5 6 based on encoder-based VLMs, following the same settings as in Tab.3, Tab.4, and Tab.5 of the main paper. These results confirm that our main conclusions remain valid even with the revised LoRA parameters. We will include these additional experiments in the revised manuscript. We hope this addresses your concerns regarding LoRA's rank ablation.
> >
> > **Q15. “Please notice the difference between data heterogeneity and client drift. Detailed discussion and references are missed. Specifically, rebuttal Q8 from the authors.“**
> >
> > Thank you for your thoughtful follow-up to Q8. We would like to clarify our understanding of your concerns and provide a detailed response. In your original question, you asked: " It's not clear whether data heterogeneity in VLMs with large amount of datasets is practical? ". Based on our understanding, the reviewers are concerned about whether data heterogeneity still exists in VLMs when each client has access to large-scale datasets. Our response aimed to explain that, in real-world settings, even with abundant local data, differences in data sources, collection conditions, and annotation standards inevitably introduce non-IID distributions, resulting in data heterogeneity across clients.
> >
> > Your follow-up question, however, "the difference between data heterogeneity and client drift. Detailed discussion and references are missed," suggests that our previous understanding may have been incomplete. We now understand that while you acknowledge the issue of data heterogeneity in federated VLM fine-tuning, your main concern is as follows: Given that VLMs have already been pretrained on large-scale datasets, does data heterogeneity among clients during federated fine-tuning still lead to client drift and consequently degrade FL performance?
> >
> > In response to this, we offer the following clarification:
> > First, we would like to clarify the differences between data heterogeneity and client drifts as suggested. Data heterogeneity refers to the non-IID nature of local datasets, which is prevalent in FL scenarios due to variations in data sources, collection methods, and label distributions (see [18]). Client drift, on the other hand, describes the divergence of local model updates from the global optimum during federated training, often caused by this very heterogeneity. The relationship between data heterogeneity and client drift is well-established in the FL literature: heterogeneity can induce client drift, leading to degraded global model performance. Numerous FL algorithms have been proposed to mitigate this, such as those using control variates for variance reduction and methods for local drift correction ([19], [20], [21]).
> >
> > Second, we would like to emphasize that our work does not aim at proposing a new algorithm to mitigate the client drift caused by data heterogeneity in federated VLM fine-tuning. Instead, as highlighted in our abstract and introduction, our main contribution is the development of FedVLMBench, the first systematic benchmark for federated fine-tuning of VLMs. Through extensive experiments, we provide valuable insights into the interplay between VLM architectures, fine-tuning strategies, data heterogeneity, and multi-task federated optimization. Importantly, during our benchmark, one of the key questions we study is whether, under the new FL setting with VLMs, data heterogeneity still induces client drift and performance drop (your posed questions), as observed in traditional FL tasks. To address this, we conduct comprehensive experiments and draw several insightful conclusions (see Tips 5 and 6), which we believe are valuable for the future practice of FL+VLM.
> >
> > We hope this will clarify your concerns.
> >
> > [18] Karimireddy S P, et al. Scaffold: Stochastic controlled averaging for federated learning. PMLR, 2020.
> >
> > [19] Gao L, et al. Feddc: Federated learning with non-iid data via local drift decoupling and correction. CVPR, 2022.
> >
> > [20] Jiang M, et al. Harmofl: Harmonizing local and global drifts in federated learning on heterogeneous medical images. AAAI, 2022.
> >
> > [21] Jothimurugesan E, et al. Federated learning under distributed concept drift. PMLR,2023.

---

> > ### Author Response · Authors · 2025-08-07
> > **Thanks for Reply**
> >
> > **Q13&Q14: The authors should provide references to support the practice of VLMs in real-world application in privacy-preserving cases.**
> >
> > Thank you for raising your concerns. We would like to further clarify the practical significance and utility of FL finetuning in VLMs from two key perspectives:
> >
> > **1. Practical need for VLM fine-tuning:**
> >
> > We would first like to address the point raised in your original review: “VLM finetuning in real world is not evidentially supported.” As noted in our response to Q10, the necessity for efficient VLM adaptation is well recognized and motivated by real-world resource constraints. Most institutions, including universities and resource-limited clinical settings, lack the computational power and data scale required for full-model training. As a result, fine-tuning pre-trained VLMs is often the only practical approach to leveraging SOTA models. This is evidenced by the widespread adoption of parameter-efficient methods (e.g., prefix tuning[1], LoRA[2]) in both general and privacy senstive domains [3-6]. Notably, LLaVA-Med[4] receives 1000+ citations, and MED-FLAMINGO[3] receives 360+ citations.
> >
> > **2. Rationale for FL in VLM Fine-tuning:**
> >
> > Given the established practicality and effectiveness of VLM finetuning, the integration of FL is a logical and increasingly necessary progression. In traditional computer vision and language models, FL is a widely recognized solution to data privacy and security concerns. For example, the classic FedAvg [12], published in 2017, has already received over 25,000 citations. Similar privacy challenges are increasingly present in VLM finetuning, especially in domains such as healthcare and finance, making the adoption of FL both relevant and timely.
> > It should be emphasized, however, that both VLM finetuning and FL+VLM are newly emerging research areas that have gained significant momentum only within the past one to two years. As such, the literature on FL+VLM is still developing and not yet as extensive or diverse as in more established fields. Nevertheless, the application of FL to VLMs has already started to attract considerable attention in recent literature even within its short timeframe, as referenced in [13-17]. A simple search for “FL” and “VLM” in the titles of ICCV 2025 accepted papers already yields numerous relevant results; we list five representative papers below [7-11]. This trend indicates growing recognition and ongoing efforts to advance this important intersection of FL and VLM.
> >
> > In summary, the necessity and feasibility of VLM finetuning—as well as its integration with FL—are strongly supported by real-world demands and a growing body of literature. Given the broad applicability of VLM finetuning and the proven value of FL in privacy-sensitive contexts, we believe that FL+VLM will become an increasingly important research direction, drawing significant attention from the community.
> >
> > BTW, references 36 (Fedllm-bench) and 37 (Openfedllm) in our paper are not duplicated citations.
> >
> > [1]Li X L, et al. Prefix-tuning: Optimizing continuous prompts for generation.arXiv.
> >
> > [2]Hu E J, et al. Lora: Low-rank adaptation of large language models. ICLR, 2022.
> >
> > [3]Moor M, et al. Med-flamingo: a multimodal medical few-shot learner. PMLR, 2023.
> >
> > [4]Li C, et al. LLaVA-Med: Training a Large Language-and-Vision Assistant for Biomedicine in One Day.NIPS, 2023.
> >
> > [5]Pan J, et al. Medvlm-r1: Incentivizing medical reasoning capability of vision-language models (vlms) via reinforcement learning. CVPR, 2025.
> >
> > [6]Alkhaldi A, et al. Minigpt-med: Large language model as a general interface for radiology diagnosis.arXiv.
> >
> > [7]Jiaqi Wu, et al. FDPT: Federated Discrete Prompt Tuning for Black-Box Visual-Language Models. ICCV, 2025.
> >
> > [8]Singha M, et al. FedMVP: Federated Multimodal Visual Prompt Tuning for Vision-Language Models. ICCV, 2025.
> >
> > [9]Miao C, et al. FedVLA: Federated Vision-Language-Action Learning with Dual Gating Mixture-of-Experts for Robotic Manipulation. ICCV, 2025.
> >
> > [10]Tastan N, et al. A Framework for Double-Blind Federated Adaptation of Foundation Models. ICCV, 2025.
> >
> > [11]Bao W, et al. Latte: Collaborative Test-Time Adaptation of Vision-Language Models in Federated Learning. ICCV, 2025.
> >
> > [12]McMahan B, et al. Communication-efficient learning of deep networks from decentralized data. PMLR, 2017.
> >
> > [13]Binqian Xu, et al. Fedmllm:Federated fine-tuning mllm on multimodal heterogeneity data. arXiv.
> >
> > [14]Jianyi Zhang, et al. Mllm-llava-fl: Multimodal large language model assisted federated learning. WACV, 2025.
> >
> > [15]Ghiasvand S, et al. pFedMMA: Personalized Federated Fine-Tuning with Multi-Modal Adapter for Vision-Language Models. arXiv.
> >
> > [16]Liao X, et al. FOCoOp: Enhancing Out-of-Distribution Robustness in Federated Prompt Learning for Vision-Language Models. ICML, 2025.
> >
> > [17]Che H, et al. Llm-driven medical report generation via communication-efficient heterogeneous federated learning. TMI, 2025.

---

> ### Comment · Reviewer_gjMf · 2025-08-08
> **Insufficient comprehensive literature discussion and involving.**
>
> The fact that practice of VLM in FL is not well supported is actually a issue of the presentation and citations. The practical application of VLM in FL is not question, but there is a lack of support in the introduction given in the submission. It is not clear in the rebuttal which parts need to be changed, how to make the promised changes, and which references need to be added.
>
> I make it further clear in details. My main concern is, **since there are so many accepted works on VLM applications in FL scenarios just on ICCV25, only two others (one of them is in arxiv) are cited in the introduciton**. Meanwhile, there are no such methods and models for selected baselines for empirical verification, as well as medical datasets.
>
> Proposing and explaining clearly is an important part of meeting NeurIPS standards for benchmark works. Due to the lack of a large number of latest references and baselines, the submitted version does not reflect the importance of VLM in FL, although many people have already studied on it.
>
> At least, these related works given by the authors should be discussed and added as baslines in the benchmark. Limited VLM FL methods are involved. **For my rating of 4 and above, the promised citations should be added and discussed, and moreover, the implementation of these baselines should be released** (no matter the authors' or the official ones).
>
> BTW, correction: references 34 and 35 are duplicated.

---

> > ### Author Response · Authors · 2025-08-08
> > **Thanks for Reply**
> >
> > **Q16: “promised changes in revision” & “only two others paper are cited in the introduction.“ & “there are no such methods and models for selected baselines for empirical verification, as well as medical datasets.”**
> >
> > We thank the reviewer for your thoughtful comments and suggestions. We are pleased to see that the reviewer shares our view on the growing popularity and importance of FL finetune of VLM. We would like to take this opportunity to further clarify several key points regarding your concerns about “insufficient comprehensive literature discussion” and to outline our revision plan.
> >
> > First, we emphasize that our work is among the earliest systematic studies at the intersection of FL and VLM, and, to the best of our knowledge, is the first to propose a dedicated benchmark for this emerging FL+VLM field. When we submitted our NeurIPS paper in May, there were very few published works directly addressing FL+VLM. The related works from ICCV 2025, ICML 2025, and Arxiv that we cited in our previous response were published after our submission. Thus, it was impractical to include them for discussion or empirical comparison in our submitted version. Actually, the rapid emergence of these concurrent works further validates the timeliness and significance of our benchmark.
> >
> > Nevertheless, we appreciate the reviewer’s suggestion to expand the discussion of related literature. We agree that incorporating a comprehensive discussion of the latest FL+VLM research will enhance our benchmark's value. We will thus make the following changes in our revision:
> >
> > **(a) Expanded Related Work:** We will update the related work section (Line 35) to include relevant FL+VLM studies, such as references [R7-R11,R13-R17] from our last response.
> >
> > **(b) Expanded Discussion of VLM Fine-tuning in Privacy-Sensitive Cases:** In the introduction (Line 31-32), we will include recent works[R3-R6] highlight the importance of federated VLM fine-tuning in privacy-sensitive domains.
> >
> > **(c) Typo Correction:** We will fix the duplicated references (34 and 35) as noted.
> >
> > For clarity, we provide the revised first paragraph of our introduction below, which addresses these points:
> >
> > “This characteristic positions VLMs as a potential foundational architecture for addressing complex open-domain tasks, not only in general domains [R18-20], but also in privacy-preserving scenarios [R3-R6]. For example, in the medical domain, several works [R3-R6] have successfully integrated visual and language information using few-shot learning [R3] or reinforcement learning [R5] to enhance the analytical capabilities of VLMs. However, existing VLM-based instruction tuning methods typically adopt a centralized learning paradigm, which fails to meet the privacy protection requirements necessary for distributed training, particularly in sensitive fields such as healthcare and finance [R3-R6]. While recent research [R7-R11,R13-R17] has introduced FL into the instruction fine-tuning of VLMs to effectively address data privacy concerns, significant limitations remain. ”
> >
> > [Note: **R** refers to the reference in previous response]
> >
> > Additionally, we would like to clarify the novelty and scope of our contribution. While the recent ICCV 2025, ICML 2025, and other concurrent works focus on methodological innovations to address specific problems within FL+VLM, our work is fundamentally different. Our main contribution is the development of FedVLMBench, the first systematic benchmark for federated fine-tuning of VLMs. Through extensive experiments, we provide valuable insights into the interplay between VLM architectures, fine-tuning strategies, data heterogeneity, and multi-task federated optimization. Our comprehensive evaluation, including additional generalization tests (such as the impact of random seeds, different VLM architectures, and various LoRA parameters), demonstrates the robustness and generality of our findings, which we believe will benefit future FL+VLM research. Furthermore, we propose and open-source six multi-modal datasets covering a variety of task types and multi-task joint fine-tuning. We have already received inquiries from researchers interested in using these datasets for their future work, underscoring the practical value and impact of our contributions.
> >
> > We hope these clarifications and planned revisions address the reviewer’s concerns and highlight the importance and originality of our work. Thank you again for your constructive feedback.
> >
> >
> > [R18] Zhai S, et al. Fine-tuning large vision-language models as decision-making agents via reinforcement learning. NIPS, 2024.
> >
> > [R19] Zhao B, et al. Tuning layernorm in attention: Towards efficient multi-modal llm finetuning. ICLR, 2024.
> >
> > [R20] Saha O, et al. Improved zero-shot classification by adapting vlms with text descriptions. CVPR, 2024.

---

> ### Comment · Reviewer_gjMf · 2025-08-09
> **Score increased to 5. Further works should be done.**
>
> I increase my rating to 5. All of the concerns are solved or promised to be in the revision. Major modification in the introduction should be done in the revision or further camera-ready.
>
> Thanks for the authors' efforts. Although the author repeatedly emphasizes that hallucinations and other VLMs beyond classification accuracy are future work, for the healthy development of FL in VLM, the very first benchmark must be comprehensive and rigorous.

---

> > ### Author Response · Authors · 2025-08-09
> > **Thanks for Reply**
> >
> > Thank you for your constructive response! Your suggestions are invaluable to us, and we will certainly incorporate your suggested modification into our revision. We are committed to continuously updating FedVLMBench, further supporting ongoing research in this area.

---

### Official Review · Reviewer_YAmn · 2025-07-03

**Rating:** 4
**Confidence:** 4

**Summary:**

This paper presents FedVLMBench, a new benchmark for evaluating federated learning (FL) methods applied to vision-language models (VLMs). The authors tackle a gap in FL research—how to effectively fine-tune VLMs in distributed, privacy-sensitive settings—by systematically testing different architectures, tuning strategies, and FL algorithms across multiple tasks.

The benchmark covers two VLM architectures (encoder-based and encoder-free), four fine-tuning approaches, five FL algorithms, and six datasets that include both single-task and multi-task scenarios. The datasets span domains like medical imaging (Fed-Med) and natural language understanding (Fed-Nature), with careful attention to data heterogeneity (IID vs. non-IID splits).

**Dataset Code Accessibility:**

Yes

**Dataset Code Comments:**

This benchmark delivers where it counts - the datasets and code are available, well-organized, and properly documented for core functionality. While I'd appreciate clearer licensing infomation and more beginner-friendly documentation like quickstart guides, these are relatively minor issues. Overall, the work provides researchers with a practical tool for evaluating federated VLMs.

**Ethical Considerations:**

No, there are no or only very minor ethics concerns

**Final Justification:**

Given that this paper provides a relatively comprehensive benchmark, it offers some value to the community. However, since it merely presents and interprets experimental results without delivering deeper insights, I am inclined to maintain my initial score.

**Limitations Weaknesses:**

1. While the benchmark includes non-IID data splits, it doesn’t test more extreme but practical challenges (e.g., straggler clients in cross-device FL or communication bottlenecks). Adding scalability experiments (e.g., varying numbers of clients) would strengthen its applicability.

2. Many results (e.g., Table 4) report averages but no variance measures. Given the randomness in FL (client sampling, initialization), confidence intervals or multiple seeds would help assess robustness.

3. The paper briefly mentions privacy but doesn’t discuss potential misuse (e.g., deploying federated VLMs in surveillance) or safeguards (e.g., differential privacy for medical datasets like Fed-Med).

**Strengths Contributions:**

1. This is a good benchmark I’ve seen that specifically targets federated fine-tuning for VLMs, moving beyond simple classification or VQA tasks to include more complex challenges like report generation and visual grounding.

2. The experiments are thorough, comparing different connector designs (e.g., linear vs. MLP adapters) and tuning strategies (e.g., sequential vs. joint tuning). The findings are actionable, such as the discovery that a 2-layer MLP connector works best for encoder-based VLMs in FL.

3. The authors provide code and dataset access, along with documentation of training setups (optimizers, hyperparameters, compute resources). This aligns well with the NeurIPS benchmark track’s emphasis on reusable tools.

---

> ### Author Rebuttal · Authors · 2025-07-31
>
> We sincerely thank the reviewer for their efforts and positive feedback. We are glad to be recognized for presenting a "good benchmark", providing detailed experiments, and delivering actionable findings. We report relevant experiment results and explanations to your comments. Please see our response below regarding the specific comments.
>
> **Q1. Adding more extreme challenges (e.g., straggler clients in cross-device FL or communication bottlenecks) and pratical experiments.**
>
> Thank you for your constructive feedback. We agree that challenges such as straggler clients and communication bottlenecks are highly relevant in practical FL scenarios, especially for fine-tuning large VLMs. However, our current work is specifically focused on systematically benchmarking parameter-efficient fine-tuning methods under data heterogeneity. This is the most common yet underexplored challenge in FL for VLMs. Due to space limitations, we do not explore other types of heterogeneity (e.g., system heterogeneity, device heterogeneity, or communication constraints), which we recognize as valuable directions for future work. We will clarify this limitation in the revised manuscript.
>
> **Q2. Given the randomness in FL (client sampling, initialization), confidence intervals or multiple seeds would help assess robustness.**
>
> Thank you for your constructive feedback. We agree that evaluating the effects of randomness is important for assessing robustness in FL. In fact, as shown in Fig. 3(a) of our supplementary material, we have already examined how different connector choices affect FL VLM performance in encoder-based VLMs across various random seeds. These results are consistent with Takeaway 1 in our paper.
>
> Following your suggestion, we further validated our most critical conclusions (Takeaways 5 and 6) using multiple random seeds.  Specifically, we conduct sensitivity experiments with encoder-free VLMs on the Fed-SLAKE, Fed-FGVC, and Fed-Nature datasets, following the same experimental settings as in Tab. 4 and Tab. 5 of the main paper. For each experiment, we use three different random seeds and reported both the mean and variance of the results.
>
> As illustrated in Tab.A6, we first measure FedAvg's performance on Fed-SLAKE and Fed-FGVC datasets. For the text-centric task (Fed-SLAKE), there is no significant difference in FedAvg's performance between IID and Non-IID distribution. However, for the vision-centric task (Fed-FGVC), we observe a considerable drop in performance under Non-IID conditions. This finding supports Takeaway **5** from our paper. We also compare the performance between centralized training (MT-Central) and FedAvg in multi-task federated fine-tuning on Fed-Nature dataset. As shown in Tab.A7, FedAvg's performance in four tasks closely matches that of centralized training, which aligns with Takeaway **6** in our paper.
>
> We will update the variance of these experimental results in the revised version.
>
>
> Table A6：Experiment results on Fed-SLAKE and Fed-FGVC datasets.
> |  Dataset |         Fed-SLAKE|     Fed-SLAKE      |   Fed-FGVC |       Fed-FGVC         |
> |:-|:-:|:-:|:-:|:-:|
> |  Method  |     IID    |     Non-IID   |     IID    |   Non-IID   |
> | Fedavg   | 0.773(4e-5)| 0.757(2.2e-5) | 0.716(5e-5)| 0.492(2e-6) |
>
> Table A7：Quantitative comparison on Fed-Nature dataset. MT-Central refers to centralized training on the centralized multi-task dataset.
>
> |    Mode    |     VQA    |   Caption Generation      | Visual Grounding| Classification|
> |:-|:-:|:-:|:-:|:-:|
> | MT-Central | 0.755(2e-5)|  0.911(5e-6)/0.365(2e-5)  |    0.461(4e-5)  |  0.876(2e-5)  |
> |   Fedavg   | 0.772(2e-4)| 0.921(1.5e-4)/0.362(1e-6) |    0.452(3e-4)  |  0.892(2e-5)  |
>
>
> **Q3. Do not include the discussion on potential misuse or safeguards.**
>
> Thank you for your valuable feedback. Our primary aim is to explore the most critical and underexplored gaps in FL for VLMs across various settings, including non-IID distributions, task types, and multi-task fine-tuning. While we agree that addressing potential misuse and proposing safeguards are very important directions, due to space limitations, we were unable to include them in this work and leave them as valuable avenues for future research.
>
>
> **Q4. More beginner-friendly documentation.**
>
> Thank you for your valuable suggestion. We have enhanced the documentation in both the code and dataset repositories to better support the research community. The updated documentation now includes: (1) an overview of the paper, (2) detailed instructions for accessing the full dataset and code, (3) guidelines for modifying configuration settings, and (4) information on experimental parameters and relevant reference repositories. We will continue to update these documents to provide a solid research foundation for researchers in the field of federated fine-tuning of VLMs.

---

> > ### Comment · Reviewer_YAmn · 2025-08-07
> > **Futher Comments**
> >
> > I sincerely appreciate the authors' responses to my concerns. Based on the current revisions, none of my remaining concerns would require additional experimental work. Considering the authors' commitment to incorporating these discussions into the manuscript, I am pleased to maintain my positive evaluation score.

---

> > > ### Author Response · Authors · 2025-08-07
> > > **Thanks for Reply**
> > >
> > > Thank you for your thoughtful follow-up! We are glad to hear that we have addressed most of your concerns regarding the experimental results and that you noted "a good benchmark I’ve seen that specifically targets federated fine-tuning for VLMs."
> > >
> > > We believe that our work has made a positive contribution to further research in the FL community. Given that you found it valuable and that most of your concerns have been addressed, we would greatly appreciate your stronger support with a potential increased score. If there are any remaining concerns, please feel free to raise them, and we would be more than happy to continue discussing them with you.

---

> ### Comment · Area_Chair_cq3X · 2025-08-05
>
> Gentle reminder. Please read through the authors' rebuttal and share any further comments. Thanks!

---

> ### Author Response · Authors · 2025-08-06
>
> We would like to kindly follow up to see if our previous response has adequately addressed all of your concerns. If there are any remaining issues or if you need further clarification, we would be more than happy to discuss with you and provide additional details and improvements. Your feedback is invaluable to us, and we truly appreciate your time and consideration.

---

### Official Review · Reviewer_3GTm · 2025-07-06

**Rating:** 4
**Confidence:** 2

**Summary:**

The paper proposes FedVLMBench for federated fine-tuning of Vision-Language Models (VLMs). FedVLMBench includes two VLM architectures (encoder-based and encoder-free), four fine-tuning strategies, five FL algorithms, and six multimodal datasets covering four single-task and two multi-task scenarios across four task categories. Extensive experiments show that the optimal configuration for encoder-based VLMs in FL  is a 2-layer MLP and the higher sensitivity of vision-centric tasks to data heterogeneity compared to text-centric ones in FL.

**Dataset Code Accessibility:**

Yes

**Dataset Code Comments:**

The author provides sufficient evidence to support reproducibility.

**Ethical Considerations:**

No, there are no or only very minor ethics concerns

**Final Justification:**

Overall, FedVLMBench provides a comprehensive evaluation that significantly contributes to research on federated learning and multimodal large language models. However, some conclusions, such as whether a two-layer MLP is truly the most effective approach in federated learning, remain unclear, which somewhat undermines its broader impact.

**Limitations Weaknesses:**

1. The author uses LLaVA 1.5’s architecture as the encoder-based VLM. However, it is not clear whether the conclusion can be applied to recent advanced VLMs like Qwen2.5-VL, Seed1.5-VL, and InternVL3.
2. Will the sensitivity of parameter initialization affect the conclusion that the 2-layer MLP is optimal?
3. From Table 4-5, we can conclude that FedAvg is superior than other FL methods. However, the paper does not address computational costs under VLM evaluation prototcal. Does the author consider the conputional cost of these methods?

**Strengths Contributions:**

1. The motivation is clear. The connector design and fine-tuning strategies are important in VLMs. FedVLMBench includes both encoder-based and encoder-free VLMs in the paper.
2. The evaluation is comprehensive. Compared with former benchmarks, FedVLMBench includes six datasets and evaluates four fine-tuning strategies and five FL algorithms. Moreover, the evaluation presents new research directions that encoder-based VLMs exhibit significant performance drops for non-IID vision-centric tasks.
3. The claims are well-supported. For more complex tasks such as report generation and visual localization, the author claims that these tasks can benefit more from connector fine-tuning than text-centric tasks. Such a conclusion is well-supported by the experiments on Fed-SLAKE and Fed-FGVC.

---

> ### Author Rebuttal · Authors · 2025-07-31
>
> We thank the reviewer for the time and effort in reviewing our paper and providing constructive comments. We are pleased that the reviewer found the paper to be ''clear-motivated'', ''comprehensive evaluation'', and ''well-supported'', which inspires us a lot. We report the suggested experiments result in our response. Please see our response below regarding the specific comments.
>
> **Q1. Can the conclusion be generalized to other advanced VLMs, such as Qwen2.5-VL, Seed1.5-VL, and InternVL3?**
>
> Thanks for your constructive suggestion. Our framework is now compatible with Qwen2.5-VL. Due to time constraints, we prioritize validating the most critical conclusions from our paper on this new model. Specifically, we conduct quantitative experiments for different FL methods on text-centric dataset (Fed-SLAKE), vision-centric dataset (Fed-FGVC), and multi-task FL dataset(Fed-Nature) for Qwen.
>
> As shown in Tab.A1, for the text-centric task (Fed-SLAKE), the performance differences between the five FL methods on Qwen2.5-VL are not significant under IID and non-IID conditions. However, for the vision-centric task (Fed-FGVC), both methods show a decrease in performance under non-IID data, with FedAvg outperforming the other methods in this setting. This observation aligns with **Takeaway 5** in our main paper.
> Tab.A2 reports the performance of different FL methods on the Fed-Nature dataset. It can be seen that federated multi-task training is close to the performance of centralized training in VQA, caption generation, and classification tasks, which is consistent with **Takeaway 6** in our paper.
>
> These experiments demonstrate that our key conclusions hold consistently across architectures (e.g., Qwen2.5-VL). We will add these valuable experiments and results to the revision.
>
> Table A1: Performance comparison of Qwen2.5-VL on two representative single-task datasets with IID and non-IID distributions.
> |   |  Fed-SLAKE |  Fed-SLAKE   |Fed-FGVC  |   Fed-FGVC    |
> |:-|:-:|:-:|:-:|:-:|
> | Method   |    IID      |   Non-IID   |    IID      |   Non-IID   |
> | Central  |    0.804    |      —      |    0.871    |      —      |
> | FedAvg   |    0.771    |    0.756    |    0.850    |    0.812    |
> | FedProx  |    0.772    |    0.761    |    0.843    |    0.765    |
> | FedAdam  |    0.760    |    0.754    |    0.831    |    0.740    |
> | FedAvgM  |    0.759    |    0.752    |    0.834    |    0.758    |
> | FedYogi  |    0.755    |    0.752    |    0.830    |    0.703    |
>
> Table A2:Performance comparison of Qwen2.5-VL on Fed-Nature dataset. MT-Central refers to centralized training on the centralized multi-task dataset.
> | Mode     |    VQA     | Caption Generation | Visual Grounding | Classification |
> |:-|:-:|:-:|:-:|:-:|
> | MT-Central |   0.784   |   1.440 / 0.446    |      0.681      |     0.975    |
> | FedAvg    |   0.777   |   1.416 / 0.440    |      0.642      |     0.973    |
> | FedProx   |   0.779   |   1.483 / 0.452    |      0.630      |     0.971    |
> | FedAdam   |   0.799   |   1.495 / 0.452    |      0.619      |     0.979    |
> | FedAvgM   |   0.774   |   1.353 / 0.433    |      0.621      |     0.970    |
> | FedYogi   |   0.793   |   1.517 / 0.453    |      0.616      |     0.977    |
>
> **Q2. Is the paper’s conclusion (e.g., the 2-layer MLP is optimal) sensitive to parameter initialization?**
>
> Due to space constraints, we have placed this experiment in the supplementary material. We will address this question in revision. As illustrated in Fig.3(a) of the supplementary material, we present the performance fluctuations of the three connectors across different random seeds. The results indicate that, under varying parameter initialization conditions, the performance and stability of the 2-layer MLP outperform that of the other connectors.
>
> In addition, we also conduct parameter initialization sensitivity experiments on Fed-SLAKE, Fed-FGVC, and Fed-Nature datasets. Due to time constraints, we prioritize validating the most critical conclusions from our paper. To verify the conclusions **Takeaway 5 6** based on encoder-free VLM, we followed the same settings as in Tab. 4 and Tab. 5 of the main paper. Specifically, we set three different random seeds for these experiments and calculated the mean and variance of all the results.
>
> Firstly, as illustrated in Tab.A3, we measured FedAvg's performance on Fed-SLAKE and Fed-FGVC datasets. For the text-centric task (Fed-SLAKE), there is no significant difference in FedAvg's performance between IID and Non-IID distribution. However, for the vision-centric task (Fed-FGVC), we observe a considerable drop in performance under Non-IID conditions. This finding supports Takeaway **5** from our paper.
> Next, we compare the performance between centralized training (MT-Central) and FedAvg in multi-task federated fine-tuning on Fed-Nature dataset. As shown in Tab.A4, FedAvg's performance in four tasks closely matches that of centralized training, which aligns with Takeaway **6** in our paper.
>
> We will add these experimental results in the revision.
>
> Table A3：Experiment results on Fed-SLAKE and Fed-FGVC datasets.
> |  Dataset |         Fed-SLAKE|     Fed-SLAKE      |   Fed-FGVC |       Fed-FGVC         |
> |:-|:-:|:-:|:-:|:-:|
> |  Method  |     IID    |     Non-IID   |     IID    |   Non-IID   |
> | Fedavg   | 0.773(4e-5)| 0.757(2.2e-5) | 0.716(5e-5)| 0.492(2e-6) |
>
> Table A4：Quantitative comparison on Fed-Nature dataset.MT-Central refers to centralized training on the centralized multi-task dataset.
>
> |    Mode    |     VQA    |   Caption Generation      | Visual Grounding| Classification|
> |:-|:-:|:-:|:-:|:-:|
> | MT-Central | 0.755(2e-5)|  0.911(5e-6)/0.365(2e-5)  |    0.461(4e-5)  |  0.876(2e-5)  |
> |   Fedavg   | 0.772(2e-4)| 0.921(1.5e-4)/0.362(1e-6) |    0.452(3e-4)  |  0.892(2e-5)  |
>
> **Q3. Concerns on computational costs under VLM evaluation protocol.**
>
> Thanks for your constructive suggestion. For fairness, we use the same LoRa parameters across all experiments (details are provided in the supplementary materials). This ensures that the number of parameters optimized by different FL methods remains consistent, with computational loss arising primarily from variations in their optimization methods.
>
> As shown in Tab.A5, we measure the average time required for different FL methods to complete a single round of communication on the Fed-Nature dataset, including client training and server-side parameter aggregation. The basic method, FedAvg, which directly aggregates parameters from different clients, employs a straightforward and efficient computational approach. FedProx regularizes the model parameters during the optimization process, which significantly increases the computational cost. This aligns with the experimental results presented in Tab.3. We will add this analysis in the revision.
>
> Table A5：The computational costs for each method on the Fed-Nature dataset for a single round of FL training.
> |  Method |# Per-round Computation Cost|
> |:-|:-:|
> | Fedavg  |          236s              |
> | Fedprox |          262s              |
> | FedAdam |          240s              |
> | FedAvgM |          241s              |
> | FedYogi |          239s              |

---

> > ### Comment · Reviewer_3GTm · 2025-08-05
> >
> > Thank you for your detailed response. I would like to keep my positive score.

---

> > > ### Author Response · Authors · 2025-08-05
> > > **Thanks for Reply**
> > >
> > > Thank you for your constructive response! We greatly appreciate your consistent support; your suggestions are invaluable to us, and we will certainly incorporate your suggested experiments into our revision. If you have any additional questions or remaining issues, please feel free to let us know. We would be pleased to discuss and explore ways to improve our work.

---

> ### Comment · Area_Chair_cq3X · 2025-08-05
>
> Gentle reminder. Please read through the authors' rebuttal and share any further comments. Thanks!

---

### Note · Authors · 2025-08-12

We sincerely thank the reviewers and the ACs for their careful reading of our paper, insightful comments and suggestions, and valuable discussions during the review process, all of which have been extremely helpful and have substantially strengthened our paper. We are pleased to note that the reviewers recognized our efforts in addressing most of the concerns raised, and we appreciate their positive evaluation of our work. Below, we summarize the common concerns, along with the revisions made in response:

**1.Generalizability of Takeaway Tips:** In response to the valuable suggestions from reviewers (3GTm, YAmn, gjMf, and ehBH), we validated the general applicability of the takeaways presented in our paper through experiments involving different VLM, randomness in FL, various LoRA parameters, zero-shot performance, and more experiments on cross-tasks datasets.

**2.Computational and Communication Costs:** Following the constructive advice of reviewers (3GTm and ehBH), we conducted experiments to analyze the computational and communication costs of various FL methods on VLMs. These experimental results align with our takeaway 5.

**3.Additional Clarifications, Limitations, and Future Work:** Informed by the insightful suggestions from reviewers (YAmn, gjMf,  and ehBH), we clarified the primary focus of our paper and the rationale behind our federated dataset design. In the revision, we will include a dedicated section to discuss the limitations of existing methods and future research directions. For example, we recognize open challenges such as device, system heterogeneity, straggler effects, hallucination, domain-aware behavior analysis, and vertical FL, among others.

**4.Strengthened Research Background:** As suggested by reviewer gjMf's constructive suggestions, we have rewritten the first paragraph of the introduction to provide a comprehensive overview of the latest FL and VLM studies, further highlighting the necessity and practicality of FL with VLM. (see the response to reviewer gjMf, Q16).

**5.Detailed Documentation:** In response to the insightful recommendations of reviewers (YAmn and gjMf), we updated the code documentation to be more beginner-friendly and plan to include new model architectures, providing a solid foundation for future researchers.

These revisions will be included in our revision. Thanks again to the reviewers for the time and effort in reviewing our paper.

---

### Decision · Program_Chairs · 2025-09-18

**Decision:**

Reject

**Comment:**

(a) Summary of Scientific Claims and Findings

This paper introduces FedVLMBench, the first comprehensive benchmark for federated fine-tuning of vision-language models (VLMs). It evaluates two major VLM architectures (encoder-based and encoder-free), four fine-tuning strategies (e.g., LoRA, MLP head tuning), and five FL algorithms across six multimodal datasets spanning both single-task and multitask cross-domain settings. Key findings include: (1) a simple 2-layer MLP connector plus LLM tuning is optimal for encoder-based VLMs in FL, and (2) vision-centric tasks suffer more from data heterogeneity than text-centric ones in federated settings.

(b) Strengths

* Timely and relevant benchmark: Addresses a growing need for privacy-preserving foundation model fine-tuning, especially in sensitive domains like healthcare.
* Breadth and realism: Covers a wide range of FL algorithms and VLM configurations with diverse downstream tasks, including cross-domain and multitask scenarios.
* Empirical insights: The analysis reveals nuanced interactions between architecture, tuning method, and task type.

(c) Weaknesses and Limitations

* Some reviewers initially questioned the generalizability of takeaways and practical relevance of the setup. While these concerns were largely resolved, the benchmark still assumes a specific formulation of FL scenarios that may not capture all real-world deployments (e.g., vertical FL or extreme device heterogeneity).
* The paper could still benefit from clearer positioning of contribution novelty relative to concurrent FL+VLM efforts.

(d) Recommendation and Justification

Accept. This paper fits the Data and Benchmark Track well, offering a new, extensible platform that systematically addresses a key gap in multimodal FL evaluation.

(e) Rebuttal and Discussion Summary

Multiple reviewers raised concerns about (1) generalizability of insights, (2) cost and efficiency analysis, (3) clarity of paper structure and motivation, and (4) limitations of existing FL methods. The authors responded with additional experiments on randomness, different LoRA settings, cost analysis, and more datasets. They rewrote the introduction, clarified goals, and added documentation updates. These changes addressed the concerns thoroughly, and reviewer sentiment improved accordingly. No major objections remain.

===== FINAL UPDATE FROM DB Track PCs ====

The final decision for this paper has been taken by the program chairs after consultation with the SACs. All Senior Area Chairs have ranked papers according to the feedback from the AC during the review process. We decided to leave the original meta-review to reflect the opinion of the AC in light of the initial discussions with reviewers and SAC.